# Modeling the role of livestock grazing in C and N cycling in grasslands with LPJmL5.0-grazing

Jens Heinke[1], Susanne Rolinski[1], and Christoph Müller[1]

[1]Potsdam Institute for Climate Impact Research (PIK), member of the Leibniz Association, P.O. Box 60 12 03, 14412 Potsdam, Germany

**Correspondence:** Jens Heinke (heinke@pik-potsdam.de)

**Abstract.** To represent the impact of grazing livestock on carbon (C) and nitrogen (N) dynamics in grasslands, we implement a livestock module into LPJmL5.0-tillage, a global vegetation and crop model with explicit representation of managed grasslands and pastures, forming LPJmL5.0-grazing. The livestock module uses lactating dairy cows as a generic representation of grazing livestock. The new module explicitly accounts for forage quality on dry matter intake and digestibility using relationships derived from compositional analyses for different forages. Partitioning of N into milk, feces, and urine are simulated by the new livestock module shows very good agreement with observation-based relationships reported in the literature. Modeled C and N dynamics depend on forage quality (C:N ratios in grazed biomass), forage quantity, livestock densities, manure or fertilizer inputs, soil, atmospheric $CO_2$ concentrations, and climate conditions. Due to the many interacting relationships, C sequestration, GHG emissions, N losses and livestock productivity show substantial variation in space and across livestock densities. The improved LPJmL5.0-grazing model can now assess the effects of livestock grazing on C and N stocks and fluxes in grasslands. It can also provide insights about the spatio-temporal variability of grassland productivity and into trade-offs between livestock production and environmental impacts.

## 1 Introduction

Grazing lands occupy about 25 % of the global land area (excluding Antarctica) (Klein Goldewijk et al., 2017) and provide nearly half of the total biomass used in global livestock production (Herrero et al., 2013). They also play an important role in Earth's carbon cycle by storing large amounts of soil organic carbon (Conant et al., 2017) and contributing to the terrestrial carbon sink (Chang et al., 2021). Grazing alters carbon and nitrogen cycling in grassland (McSherry and Ritchie, 2013; Conant et al., 2017; He et al., 2020; Zhou et al., 2017) with potential effects on carbon stocks, carbon uptake, $N_2O$ emissions, $NO_3^-$ leaching, and $NH_3$ volatilization. In addition, grazing ruminants produce large amounts of $CH_4$, a potent greenhouse gas (GHG).

Dynamic global vegetation models (DGVMs) are powerful tools to quantify global biogeochemical cycles, including important aspects such as carbon uptake and loss from the terrestrial biosphere and their response to changing climate and management (Fernández-Martínez et al., 2019; Kondo et al., 2020). Yet, most state-of-the-art DGVMs do not account for grazing or only in a simplified form by removing a fixed fraction of the above-ground living biomass and, in some cases, returning

part of it to grassland litter pools (Friedlingstein et al., 2022). A notable exception is ORCHIDEE-GM (Chang et al., 2013), which employs the animal module of the process-based grassland model PaSIM (Riedo et al., 1998; Vuichard et al., 2007) to estimate livestock feed intake from stocking density, animal weight and biomass availability, and to determine the partitioning of C and N into maintenance respiration, products, feces, and urine. However, the effect of feed composition on feed intake and digestibility of C, N, and energy in feed are not accounted for.

In this paper, we describe the implementation of grazing livestock – represented by lactating dairy cows – into the dynamic global vegetation and agricultural model LPJmL5-tillage (Lutz et al., 2019), forming LPJmL5.0-grazing. LPJmL5-tillage incorporates an explicit representation of the N cycle (von Bloh et al., 2018) and includes all improvements of grassland vegetation and carbon dynamics described in Rolinski et al. (2018). The representation of lactating dairy cows is primarily based on established relationships also used in livestock management applications (National Research Council, 2001) supplemented by relationships from the scientific literature. To account for the effect of forage quality on the digestibility of C and N, as well as digestible energy content, we use compositional data for a wide range of forage plants from Feedipedia (2020) to link these properties to forage N content.

## 2   Model description

The principal concept of the new livestock module is to determine the mass balances of C and N for grazing dairy cows. Both arranged in a way that one term can be estimated as the remainder of the balance. In the N balance that remainder is urinary N excretion ($m_{\mathrm{N,urine}}$):

$$m_{\mathrm{N,urine}} = m_{\mathrm{N,intake}} - m_{\mathrm{N,feces}} - m_{\mathrm{N,milk}} \tag{1}$$

where $m_{\mathrm{N,intake}}$ (Eq. 29) is N intake with forage, $m_{\mathrm{N,feces}}$ is N excreted with feces (Eq. 48), and $m_{\mathrm{N,milk}}$ is N in milk produced (Eq. 42). In the C balance, the respective remainder is C converted to $CO_2$ through respiration ($m_{\mathrm{C,respiration}}$):

$$m_{\mathrm{C,respiration}} = m_{\mathrm{C,intake}} - m_{\mathrm{C,feces}} - m_{\mathrm{C,urine}} - m_{\mathrm{C,methane}} - m_{\mathrm{C,milk}} \tag{2}$$

where $m_{\mathrm{C,intake}}$ is C intake with forage (Eq. 30), $m_{\mathrm{C,feces}}$ is C excreted with feces (Eq. 49), $m_{\mathrm{C,urine}}$ is C excreted with urine (Eq. 50), $m_{\mathrm{C,methane}}$ is C in methane from enteric fermentation (Eq. 47), and $m_{\mathrm{C,milk}}$ is C in milk produced (Eq. 43).

All terms on the right-hand side of Eqs. 1 and 2 depend directly or indirectly on forage composition, i.e, the mass fractions of crude protein, fatty acids, non-fiber carbohydrates (starch and sugars), fiber carbohydrates (cellulose and hemicellulose), and lignin. However, the only constituents of biomass in LPJmL5.0-grazing are C and N, so that all relevant forage properties that depend on forage composition need to be linked to a metric based on C and N content. We use compositional data for a wide range of forage plants from Feedipedia (2020) to calculate the weight fraction of C in forage dry matter and the weight fraction of N in the total mass of C and N ($w_{\mathrm{C,DM}}$ and $w_{\mathrm{N,CN}}$, respectively; section 2.1), digestible fractions of C and N in

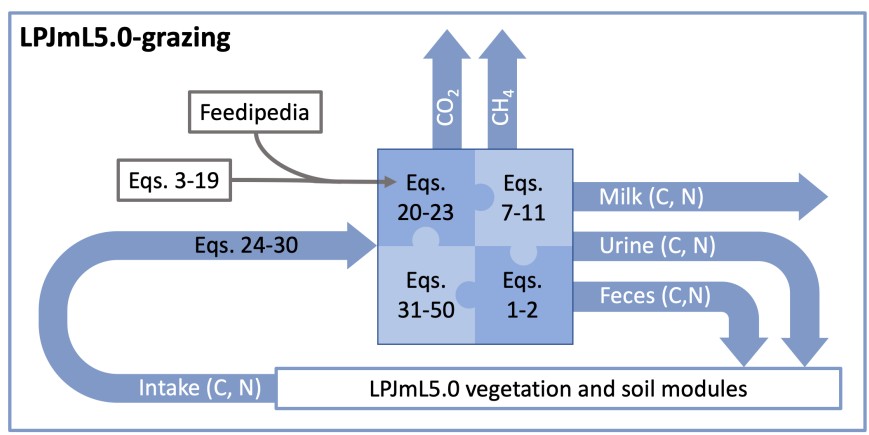

**Figure 1.** Schematic representation of LPJmL5.0-grazing. The new livestock module calculates intake of C and N from grazing and determines their partitioning into $CO_2$, methane ($CH_4$), and C and N in milk, urine, and feces. The equations described in the method section can be broadly divided into the following groups: Eqs. 3–19 are used to process compositional data from Feedipedia (2020) to derive Eqs. 20–23, which link important variables required in the main model equations (Eqs. 31–50) to N in forage ($w_{\mathrm{N,CN}}$). Eqs. 26–30 are used to calculate C and N intake, Eqs. 1–2 are the mass balance equations of N and C, respectively, and Eqs. 7–10 are weight fractions of C and N in protein, fat, and carbohydrates used to calculate the C and N content of milk.

forage ($f_C$ and $f_N$; section 2.2), and the net energy content of forage dry matter ($ne$, section 2.3). We then determine statistical relationships for $f_C$, $f_N$, $ne$, and $w_{\mathrm{C,DM}}$ in relation to $w_{\mathrm{N,CN}}$ (section 2.4) to calculate these parameters in LPJmL5.0-grazing.

The remaining part of the method section is structured as follows: Section 2.5 describes the calculation of dry matter intake, section 2.6 the calculation of energy and protein requirements of livestock, section 2.7 the calculation of milk production, section 2.9 the calculation of C and N in feces and urine, and section 2.8 the calculation of methane emissions from enteric fermentation. Fig. 1 provides a schematic representation of the LPJmL5.0-grazing with the new livestock module and its linkages to LPJmL5.0 through fluxes of C and N. In addition, we provide an R script with a fully functional implementation of the livestock model as supplementary material.

## 2.1 Conversion of forage dry matter to C and N

### 2.1.1 Composition of forage dry matter

Compositional analyses from Feedipedia (2020) provide mass fractions of forage dry matter for crude protein ($w_{\mathrm{CP,DM}}$), neutral detergent fibre ($w_{\mathrm{NDF,DM}}$), lignin ($w_{\mathrm{L,DM}}$), ether extract ($w_{\mathrm{EE,DM}}$), and ash ($w_{\mathrm{A,DM}}$). $w_{\mathrm{NDF,DM}}$ contains a small amount of neutral detergent insoluble crude protein (NDICP), which is also included in $w_{\mathrm{CP,DM}}$ and needs to be subtracted to obtain the mass fraction of nitrogen-adjusted NDF in forage dry matter (National Research Council, 2001):

$$w_{\mathrm{nNDF,DM}} = w_{\mathrm{NDF,DM}} - w_{\mathrm{NDICP,DM}} \tag{3}$$

where $w_{\mathrm{NDICP,DM}}$ is the weight fraction of NDICP in forage dry matter. From data for a wide range of grasses and legumes, Weiss et al. (1992) determined the following relationship for the estimation of the weight fraction of NDICP in forage dry matter:

$$w_{\mathrm{NDICP,DM}} = -0.0877 + 0.33 \cdot w_{\mathrm{CP,DM}} + 0.143 \cdot w_{\mathrm{NDF,DM}} \tag{4}$$

From the mass fractions of crude protein, nitrogen-adjusted neutral detergent fibre, ether extract, and ash, the mass fraction of non-fiber carbohydrates is determined as a residuum (National Research Council, 2001):

$$w_{\mathrm{NFC,DM}} = 1 - w_{\mathrm{CP,DM}} - w_{\mathrm{nNDF,DM}} - w_{\mathrm{EE,DM}} - w_{\mathrm{A,DM}} \tag{5}$$

The ether extract obtained from nutritional analysis consists of all lipids in the sample. Some of these lipids, such as pigments and waxes, have almost no nutritional value (Weiss et al., 1992). To obtain the weight fraction of highly digestible fatty acids, $w_{\mathrm{EE,DM}}$ needs to be adjusted to account for about 1 % non-fatty acids components in forage dry matter (National Research Council, 2001):

$$w_{\mathrm{FA,DM}} = \begin{cases} w_{\mathrm{EE,DM}} - 0.01 & \text{if } w_{\mathrm{EE,DM}} > 0.01 \\ 0 & \text{otherwise} \end{cases} \tag{6}$$

### 2.1.2 C and N content of crude protein

Proteins are large macromolecules, which consist of amino acids linked by peptide bonds. There are 20 commonly occurring proteinogenic amino acids with molar weights between 75.1 and 204.2 g mol$^{-1}$, mass fractions of C between 0.30 and 0.65, and mass fractions of N between 0.08 and 0.32. When amino acids are linked through peptide bonds, one molecule of water is released for each bond. Therefore, anhydrous amino acids in peptide chains have a molar weight that is about 18 g mol$^{-1}$ lower and a higher mass fraction of C and N of 0.35 to 0.73 and 0.09 to 0.36, respectively.

The C and N content of proteins varies depending on their amino acid composition. Tomé et al. (2019) give a range of 13 to 19 % for the N content of proteins, but data on C content are rare. Kozlowski (2017) analyzed nearly 14 billion amino acids in nearly 30 million proteins to determine the composition of proteins in 1612 eucaryote species. Using the average amino acid frequencies across all eucaryote species, we estimate a C fraction of 0.53 and a N fraction of 0.17 for average eucaryote protein. Using the amino acid frequencies for the two grass species in the database (Setaria viridis and Eragrostis curvula), we estimate a C fraction of 0.53 and a N fraction of 0.18 for protein in both grass species. Given the close agreement of estimated C fractions, we assume for the weight fraction of C in crude protein:

$$w_{\mathrm{C,CP}} = 0.53 \tag{7}$$

The estimates of N content, however, are different and both larger than the N fraction of 0.16 assumed in compositional analyses (Santos and Huber, 2002; Feedipedia, 2020). Because all estimates of crude protein have been calculated from measured N

content using that lower value, we also use it to convert crude protein back to N. Thus, the weight fraction of N in crude protein is defined as:

$$w_{\mathrm{N,CP}} = 0.16 \tag{8}$$

### 2.1.3 C content of carbohydrates

The basic building blocks of carbohydrates are hexoses ($C_6H_{12}O_6$) and pentoses ($C_5H_{10}O_5$) with a C fraction of 0.40 each. In polymers, their monomers (hexosans and pentosans) are linked by glycosidic bonds, which are formed by releasing one water molecule for each bond. Therefore, polysaccharides from hexoses and pentoses have a higher C fraction of 0.44 and 0.45, respectively. Disaccharides and oligosaccharides lie in-between.

Carbohydrates in plant fiber comprise mainly cellulose and hemicellulose. Cellulose is a polymer of glucose (a hexose), hemicellulose consists of a mix of hexosans and pentosans (Abu Ghalia and Dahman, 2017). Information on the composition of non-fiber carbohydrates is scarce but it can be assumed to be a mix of starch (a polymer of glucose) and various disaccharides and oligosaccharides, which allows for a relatively broad range of possible C fractions. However, the mass fraction of non-fiber carbohydrates in total carbohydrates in the data from Feedipedia (2020) is small (in average 17 %), which limits its relevance for the estimation of C content in forage dry matter. Therefore, we assume the C fraction of starch and celluloses as representative value for the weight fraction of all carbohydrates:

$$w_{\mathrm{C,CHO}} = 0.44 \tag{9}$$

### 2.1.4 C content of fatty acids

The mass fraction of C of fatty acids primarily depends on their chain length. Caprylic acid, a saturated fatty acid with eight C atoms ($C_8H_{16}O_2$), has a C fraction of 0.67, while stearic acid, saturated fatty acid with 18 C atoms ($C_{18}H_{36}O_2$), has a C fraction of 0.76. The degree of saturation also has an effect on C content. For example, linolenic acid, a triunsaturated fatty acid with 18 C atoms ($C_{18}H_{30}O_2$), has a C fraction of 0.78. Triglycerides (esters from glycerol and three fatty acids) have a higher C content than single fatty acids of the same type, but the effect becomes negligible with increasing chain length. The triglyceride of linolenic acids has the same C content of 0.78 as linolenic acid.

The composition of ether extract is not reported in the data from Feedipedia (2020). According to a meta-analysis by Glasser et al. (2013), linolenic acid is the by far most abundant fatty acid in grasses, making up more than half of total lipids. Other important fatty acids are linoleic acid and palmitic acid, each contributing about 10 to 20 % to total lipids. The C content of these two fatty acids is slightly lower with 0.77 and 0.75, respectively. Because of the dominance of linolenic acid and the similar C content of other fatty acids we assume the C fraction of linolenic acid for the weight fraction of all fats and fatty acids:

$$w_{\mathrm{C,FA}} = 0.78 \tag{10}$$

### 2.1.5 C content of lignin

Lignin is a group of large macromolecules derived mainly from three precursors: p-coumaryl alcohol, coniferyl alcohol, and sinapyl alcohol (Amthor, 2003). During polymerization, the monomer units hydroxyphenyl (H), guaiacyl (G), and syringyl (S) are formed from these alcohols, which crosslink through a variety of bonds to form complex three-dimensional macro-molecules. The H, G, and S units in lignin have molecular weights of 149.2, 179.2, and 209.2 g mol$^{-1}$ and C fractions of 0.73, 0.67, and 0.63 (Amthor, 2003).

The abundance of H, G, and S units in lignin varies depending on species and tissue type. We obtain the monomer composition for lignin from eight different herbaceous plants from Baucher et al. (1998) and determine their C fraction in lignin using molecular weight and C fractions of the three monomers. Despite considerable differences in composition, we find all C fractions to be between 0.65 and 0.67 with an overall average of 0.66. Based on that we define the weight fraction of C in lignin as:

$$w_{C,L} = 0.66 \tag{11}$$

### 2.1.6 C and N content of dry matter

The C content of forage dry matter is calculated from the different forage components multiplied by their respective C fraction:

$$w_{C,DM} = w_{C,CP} \cdot w_{CP,DM} + w_{C,CHO} \cdot (w_{NFC,DM} + w_{nNDF,DM} - w_{L,DM}) + w_{C,L} \cdot w_{L,DM} + w_{C,FA} \cdot w_{EE,DM} \tag{12}$$

Note that for the calculation of total C content the weight fraction of total lipids (ether extract) is used, which also comprises pigments and waxes.

The N content in forage dry matter is calculated from crude protein only:

$$w_{N,DM} = w_{N,CP} \cdot w_{CP,DM} \tag{13}$$

We also calculate the fraction of N in the sum of C and N:

$$w_{N,CN} = \frac{w_{N,DM}}{w_{C,DM} + w_{N,DM}} \tag{14}$$

This variable can also be calculated in LPJmL5.0-grazing, where biomass is represented in terms of C and N only.

## 2.2 Digestible nutrients

The nutritious value of forage components does not only depend on their energy content but also how well they can be digested. Weight fractions of digestible nutrients ($d$) in total dry matter from NFC, CP, FA, and nNDF are calculated using relationships from National Research Council (2001):

$$d_{NFC} = 0.98 \cdot w_{NFC,DM} \tag{15a}$$

$$d_{\text{CP}} = \exp\left(-1.2 \cdot \frac{w_{\text{ADICP,DM}}}{w_{\text{CP,DM}}}\right) \cdot w_{\text{CP,DM}} \tag{15b}$$

$$d_{\text{FA}} = w_{\text{FA,DM}} \tag{15c}$$

$$d_{\text{nNDF}} = 0.75 \cdot \left[1 - \left(\frac{w_{\text{L,DM}}}{w_{\text{nNDF,DM}}}\right)^{0.667}\right] \cdot (w_{\text{nNDF,DM}} - w_{\text{L,DM}}) \tag{15d}$$

Ash and lignin do not contribute digestible nutrients. $w_{\text{ADICP,DM}}$ in Eq. 15b is the mass fraction of acid detergent insoluble crude protein (ADICP) in forage dry matter, which is estimated from $w_{\text{NDICP,DM}}$ using a relationship from Clipes et al. (2006):

$$d_{\text{ADICP}} = 0.008145 + 0.1131 \cdot w_{\text{NDICP,DM}} \tag{16}$$

For the calculations within LPJmL5.0-grazing, the fraction of digestible C from the total C in dry matter is required, which is calculated as:

$$f_{\text{C}} = \frac{w_{\text{C,CP}} \cdot d_{\text{CP}} + w_{\text{C,CHO}} \cdot (d_{\text{NFC}} + d_{\text{nNDF}}) + w_{\text{C,FA}} \cdot w_{\text{FA,DM}}}{w_{\text{C,DM}}} \tag{17}$$

Similarly, the fraction of digestible N from the total N in dry matter is calculated as:

$$f_{\text{N}} = \frac{w_{\text{N,CP}} \cdot d_{\text{CP}}}{w_{\text{N,DM}}} \tag{18}$$

## 2.3 Energy value of forages

The digestible energy $de$ in $\text{Mcal kg}^{-1}$ of forage dry matter is estimated by multiplying the mass fractions of digestible forage components with their energy content (National Research Council, 2001):

$$de = 4.2 \cdot d_{\text{NFC}} + 5.6 \cdot d_{\text{CP}} + 9.4 \cdot d_{\text{FA}} + 4.2 \cdot d_{\text{nNDF}} - 0.3 \tag{19}$$

Heat of combustion is $4.2\,\text{Mcal kg}^{-1}$ for carbohydrates, $5.6\,\text{Mcal kg}^{-1}$ for protein, and $9.4\,\text{Mcal kg}^{-1}$ for long chain fatty acids (National Research Council, 2001). Because Eqs. 15a, 15b, 15c, and 15d give true digestibilities, a correction for metabolic fecal energy is needed, which is assumed as $0.3\,\text{Mcal kg}^{-1}$ of forage dry matter (National Research Council, 2001). In a strict sense, this calculation of $de$ according to Eq. 19 is valid for energy intake at maintenance only. At higher levels of forage intake, the digestibility of diets containing high shares of digestible nutrients (above 60 %) is reduced (National Research Council, 2001). While this is relevant in highly productive dairy systems, where intake can exceed the maintenance level by a factor of four, the intake above maintenance in grazing systems is only moderate (see Fig. 3). In addition, the share of digestible nutrients in forages from Feedipedia (2020) is 54 % on average and rarely exceeds 60 %. We therefore assume that this effect is negligible for grazing cattle and do not apply a correction for intake above maintenance.

## 2.4 Forage properties in relation to $w_{\text{N,CN}}$

For the application within LPJmL5.0-grazing, we calculate $f_{\text{C}}$, $f_{\text{N}}$, $ne$, and $w_{\text{C,DM}}$ for a wider range of forage plants from Feedipedia (2020) and determine their relationship with $w_{\text{N,CN}}$ (Fig. 2). We test three different types of functional relation-

ships: (i) no relationship with $w_{N,CN}$ (i.e., the mean of the dependent variable), (ii) a linear relationship, and (iii) a non-linear exponential relationship with three parameters. All three relationships are determined for each forage property and the relationship with the lowest value for the Akaike's information criterion (AIC) is selected.

For the digestibility of C ($f_C$), we find the lowest AIC for a linear relationship with $w_{N,CN}$:

$$f_C = 0.561 + 2.190 \cdot w_{N,CN} \tag{20}$$

The estimated parameters are statistically significant with $p$-values $< 0.001$. Residual standard error (RSE) of the fitted model about 0.036, which translates into a 95 % prediction interval of $\pm 10.8$ % at $w_{N,CN} = 0.05$.

For the digestibility of N ($f_N$), we find the lowest AIC for a concave exponential relationship with $w_{N,CN}$:

$$f_N = 0.914 - 0.494 \cdot \exp\left(-59.559 \cdot w_{N,CN}\right) \tag{21}$$

All estimated parameters are statistically significant with $p$-values $< 0.001$. RSE is small (0.0078), which translates into a narrow 95 % prediction interval of $\pm 1.8$ % at $w_{N,CN} = 0.05$.

Similar to digestibility of C, digestible energy in forage ($de$) increases linearly with $w_{N,CN}$:

$$de = 1.952 + 11.438 \cdot w_{N,CN} \tag{22}$$

Also here, the estimated parameters are statistically significant with $p$-values $< 0.001$. Similar to the model of C digestibility, the RSE is relatively large (0.15), which results in a similar 95 % prediction interval of $\pm 12.0$ % at $w_{N,CN} = 0.05$.

C content of forage dry matter ($w_{C,DM}$) is found to be independent of $w_{N,CN}$:

$$w_{C,DM} = 0.424 \tag{23}$$

However, sample standard deviation of $w_{C,DM}$ is only 0.01, which translates into a 95 % prediction interval of $\pm 4.7$ %.

## 2.5 Dry matter intake

Relationships for predicting dry matter intake of lactating dairy cows usually include milk production as an independent variable because their purpose is to determine the amount and composition of feed required to achieve a desired milk yield (National Research Council, 2001). To meet the nutritional requirements of high yielding dairy cows, their diets must contain high proportions of concentrates (e.g., maize and barley) to achieve a high concentration of readily available nutrients. Such relationships are obviously inappropriate for the estimation of voluntary forage intake of grazing cows, which is limited by the capacity of the rumen to digest fibrous materials rather than the metabolic capacity of the animal to utilize the available energy (Tedeschi et al., 2019). Several relationships for predicting ad libitum forage intake of grazing cattle have been proposed but most of them have not been explicitly developed for lactating cows. Forage intake of lactating cows is higher than for dry cows of equal size because lactation causes an increase in size of the gastrointestinal tract (Coleman et al., 2014). Tulloh (1966) has measured the size of the gastrointestinal tract of twin pairs of lactating and dry cows and found that the weight of the whole

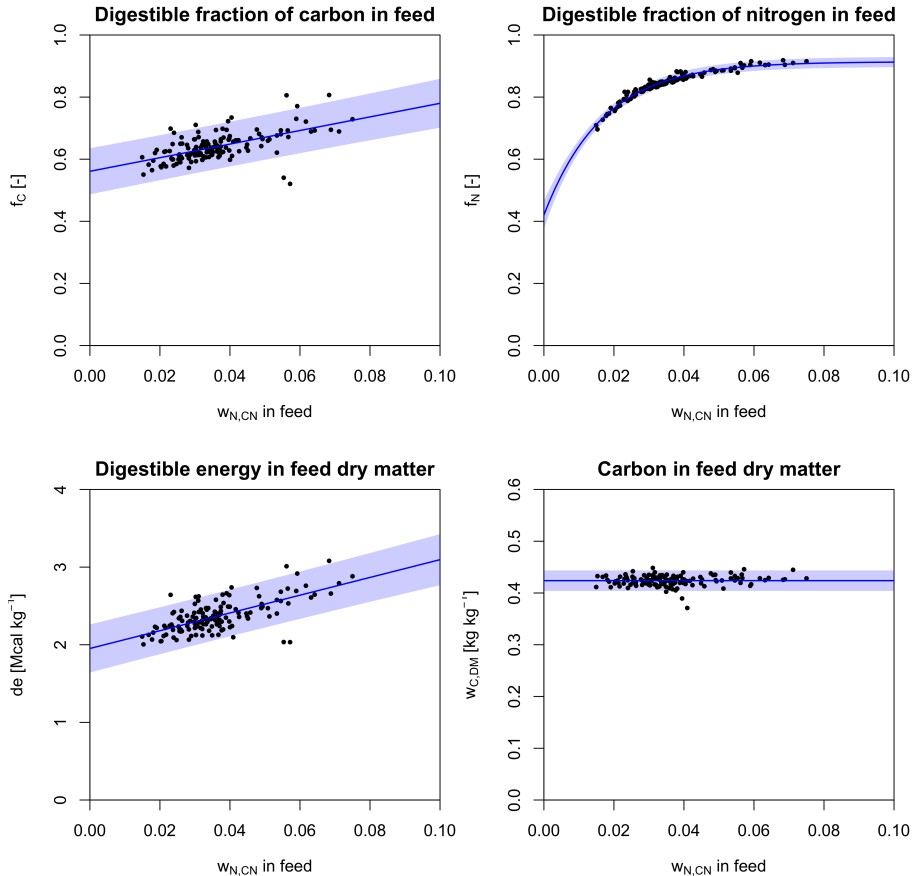

**Figure 2.** Relationships between forage properties and $w_{\mathrm{N,CN}}$ derived from data from Feedipedia (2020). Shaded areas denote 95 % prediction intervals.

tract per body weight was in average 33.1 % higher in lactating cows. The weight and the water-filled volume of the reticulo-rumen per body weight were in average about 21.3 % and 44.6 % higher in lactating cows. We choose a relationship from
Coleman et al. (2014) obtained from data of Moore et al. (1999) that predicts voluntary forage intake per kg of body weight as a function of crude protein content in forages. Because the relationship was derived from data of dry dairy cows, we multiply body weight by 1.33 for the calculation of total daily dry matter intake ($DMI_{max}$) to account for the larger gastrointestinal tract of lactating cows:

$$DMI_{max} = 1.33 \cdot BW \cdot [0.0235 - 0.0385 \cdot \exp\left(-32 \cdot w_{\mathrm{CP,DM}}\right)] \tag{24}$$

where $DMI_{max}$ is in $\mathrm{kg\,d^{-1}}$, $BW$ is the body weight in kg, and $w_{\mathrm{CP,DM}}$ is mass fraction of crude protein in forage dry matter. The equation predicts a maximum daily forage intake of 2.82 % of body weight or $14.1\,\mathrm{kg\,d^{-1}}$ dry matter for a lactating cow

weighing 500 kg. At low levels of crude protein in the feed ($w_{CP,DM} < 0.10$), the ability of ruminal microbes to break down fibrous material is affected by low N availability, which leads to lower passage rates and lower feed intake (Coleman, 2005).

The protein content in forage dry matter ($w_{CP,DM}$) required in Eq. 24 for the estimation of $DMI_{max}$ is calculated by dividing N content of forage dry matter ($w_{N,DM}$) by the N content of CP (see section 2.1.2):

$$w_{CP,DM} = \frac{w_{N,DM}}{w_{N,CP}} \tag{25}$$

$w_{N,DM}$ can be calculated from $w_{N,CN}$ by rearranging Eq. 14:

$$w_{N,DM} = \frac{w_{C,DM} \cdot w_{N,CN}}{1 - w_{N,CN}} \tag{26}$$

We note that an alternative model for estimating voluntary feed intake of dairy cows has been proposed by Faverdin et al. (2011). Unlike our approach, which is based on the NRC energy and protein system (National Research Council, 2001), their model is based on the INRA fill unit system (Institut national de la recherche agronomique, 2007) and requires iteration between multiple equations of that system. Therefore, it is not applicable here.

The dry matter intake calculated in Eq. 24 only considers limitations by the animal and does not account for the effect of biomass availability. When biomass availability declines, the grazing efficiency of the animals is reduced, because less biomass is acquired with each bite (Hodgson, 1985). To account for this effect, we adopt the sigmoid grazing function used in the Hurley model (Johnson and Parsons, 1985), which describes the decline in the proportion of $DMI_{max}$ as a function of leaf area index (LAI):

$$DMI = DMI_{max} \cdot \frac{(LAI/K)^q}{1 + (LAI/K)^q} \tag{27}$$

where $q$ is a parameter that determines the slope of the curve, and $K$ a position parameter that can be interpreted as the LAI value at which half of $DMI_{max}$ is achieved. For $q$, a value of 3 is suggested to give realistic results (Johnson and Parsons, 1985; Herrero et al., 2000). For the calculation of $K$, we adapt a relationship from Herrero et al. (2000), which accounts for the effect of animal size (body weight):

$$K = 0.229 \cdot BW^{0.36} \tag{28}$$

For a cow of 500 kg, a value of 2.15 for $K$ is obtained.

The daily intake of N per head ($m_{N,intake}$) is calculated by multiplying $DMI$ with the N content of forage dry matter:

$$m_{N,intake} = DMI \cdot w_{N,DM} \tag{29}$$

The daily intake of C per head ($m_{C,intake}$) is calculated by multiplying $DMI$ with the C content of forage dry matter:

$$m_{C,intake} = DMI \cdot w_{C,DM} \tag{30}$$

## 2.6 Energy and protein requirements

Calculations of energy requirements in National Research Council (2001) are based on net energy units, which represent energy available to the animal after all losses have been subtracted. Part of the digestible energy are lost as urine and methane, which reduces the amount of energy that is actually metabolized by the body (Weiss, 2011). Metabolizable energy $me$ in $\mathrm{Mcal\,kg^{-1}}$ of forage dry matter is calculated from $de$ (National Research Council, 2001):

$$me = 1.01 \cdot de - 0.45 \tag{31}$$

Part of the metabolizable energy is lost as heat. Net energy $ne$ in $\mathrm{Mcal\,kg^{-1}}$ of forage dry matter that is available for maintenance, activity, and lactation is calculated as (National Research Council, 2001):

$$ne = 0.703 \cdot me - 0.19 \tag{32}$$

Over a range for $w_{\mathrm{N,CN}}$ from 0.02 to 0.1, the ratio of $ne$ to $de$ increases from 50 % to 57 %.

Daily net energy requirements for maintenance are proportional to metabolic body weight $BW^{0.75}$ (National Research Council, 2001):

$$NE_{\mathrm{M}} = 0.08 \cdot BW^{0.75} \tag{33}$$

The value of $0.08\,\mathrm{Mcal\,day^{-1}\,kg^{-0.75}}$ metabolic body weight accounts for the increased maintenance requirements of lactating cows and includes a 10 % activity allowance (National Research Council, 2001). Although additional energy requirements for grazing and walking are not included in this allowance (National Research Council, 2001), they are ignored here and will be addressed in a subsequent version of our model.

Requirements of metabolizable protein ($MP$) consist of urinary protein requirements ($MP_{\mathrm{UP}}$) and metabolic fecal protein ($MP_{\mathrm{MFP}}$). $MP_{\mathrm{UP}}$ is proportional to $BW^{0.5}$ (National Research Council, 2001):

$$MP_{\mathrm{UP}} = 0.0041 \cdot BW^{0.5} \tag{34}$$

whereas $MP_{\mathrm{MFP}}$ is proportional to $DMI$ (National Research Council, 2001):

$$MP_{\mathrm{MFP}} = 0.03 \cdot DMI \tag{35}$$

## 2.7 Milk production

Net energy requirements for milk production (lactation) equal the energy content of milk (National Research Council, 2001):

$$NE_{\mathrm{milk}} = 0.36 + 9.69 \cdot w_{\mathrm{fat,milk}} \tag{36}$$

where $w_{\mathrm{fat,milk}}$ is the weight fraction of fat in milk. We assume a constant fat content of 4 % in milk ($w_{\mathrm{fat,milk}} = 0.04$), which corresponds to net energy requirements for milk production of $0.748\,\mathrm{Mcal\,kg^{-1}}$.

The amount of milk that can be produced from the available net energy above maintenance requirements is calculated as:

$$m_{\mathrm{milk,NE}} = \frac{DMI \cdot ne - NE_{\mathrm{M}}}{NE_{\mathrm{milk}}} \tag{37}$$

Requirements of metabolizable protein for lactation are calculated from milk protein content assuming a conversion efficiency of metabolizable protein to milk protein of 0.67 (National Research Council, 2001):

$$MP_{\mathrm{milk}} = \frac{w_{\mathrm{protein,milk}}}{0.67} \tag{38}$$

where $w_{\mathrm{protein,milk}}$ is the weight fraction of protein in milk. Assuming a constant protein content of 3.2 % in milk ($w_{\mathrm{protein,milk}} = 0.032$) gives metabolizable protein requirements for milk production of 0.048 kg kg$^{-1}$.

The amount of milk that can be produced from the available $MP$ above maintenance is calculated as:

$$m_{\mathrm{milk,MP}} = \frac{MP_{\mathrm{avl}} - MP_{\mathrm{UP}} - MP_{\mathrm{MFP}}}{MP_{\mathrm{milk}}} \tag{39}$$

where $MP_{avl}$ is total available metabolizable protein calculated from digested N:

$$MP_{avl} = \frac{DMI \cdot w_{\mathrm{N,DM}} \cdot f_{\mathrm{N}}}{w_{\mathrm{N,CP}}} \tag{40}$$

The actual amount of milk that can be produced is the minimum of $m_{\mathrm{milk,NE}}$ and $m_{\mathrm{milk,MP}}$:

$$m_{\mathrm{milk}} = \min(m_{\mathrm{milk,NE}}, m_{\mathrm{milk,MP}}) \tag{41}$$

The amount of N contained in milk ($m_{\mathrm{N,milk}}$) are calculated by multiplying with the mass fraction of N in milk ($w_{\mathrm{N,milk}}$):

$$m_{\mathrm{N,milk}} = w_{\mathrm{N,milk}} \cdot m_{\mathrm{milk}} \tag{42}$$

In the same way, the amount of C is determined:

$$m_{\mathrm{C,milk}} = w_{\mathrm{C,milk}} \cdot m_{\mathrm{milk}} \tag{43}$$

Weight fractions of N ($w_{\mathrm{N,milk}}$) and C $w_{\mathrm{C,milk}}$ in milk can be determined from milk composition:

$$w_{\mathrm{N,milk}} = w_{\mathrm{N,CP}} \cdot w_{\mathrm{protein,milk}} \tag{44}$$

and

$$w_{\mathrm{C,milk}} = w_{\mathrm{C,CP}} \cdot w_{\mathrm{protein,milk}} + w_{\mathrm{C,CHO}} \cdot w_{\mathrm{CHO,milk}} + w_{\mathrm{C,FA}} \cdot w_{\mathrm{fat,milk}} \tag{45}$$

Assuming a constant composition of 4 % fat, 3.2 % protein, 4.85 % sugar in milk, we obtain estimates for $w_{\mathrm{N,milk}}$ and $w_{\mathrm{C,milk}}$ of 0.00512 and 0.0695, respectively.

## 2.8 Methane from enteric fermentation

Enteric fermentation in the rumen produces methane as a by-product. According to IPCC guidelines for greenhouse gas inventories, $6.5\% \pm 1.0\%$ of gross energy intake is converted to methane (IPCC, 2006). The lower and upper bound of this range are described to be appropriate for "good" and "poorer" feed, respectively, but a quantitative relationship is not given by IPCC (2006). Hence, we calculate methane production ($m_{\mathrm{methane}}$ in $\mathrm{kg\,day^{-1}}$) assuming a constant methane conversion factor of $6.5\%$ of gross energy intake:

$$m_{\mathrm{methane}} = \frac{DMI \cdot 18.4 \cdot 0.065}{55.6} \tag{46}$$

where 18.4 and 55.6 are the gross energy content of feed and methane, respectively, in $\mathrm{MJ\,kg^{-1}}$. Note that in light of the large uncertainty entailed with assuming a constant emission factor, we do not account for possible variations in gross energy content for forage here.

Since C makes up $75\%$ of the molar weight of methane ($12\,\mathrm{g\,mol^{-1}}$ out of $16\,\mathrm{g\,mol^{-1}}$), the amount of C converted to methane is calculated as:

$$m_{\mathrm{C,methane}} = 0.75 \cdot m_{\mathrm{methane}} \tag{47}$$

## 2.9 Feces and urine

N excreted with feces comprises indigestible N in feed and the N contained in metabolic fecal protein:

$$m_{\mathrm{N,feces}} = m_{\mathrm{N,intake}} \cdot (1 - f_{\mathrm{N}}) + w_{\mathrm{N,CP}} \cdot MP_{\mathrm{MFP}} \tag{48}$$

Analogously, C excreted with feces is calculated as:

$$m_{\mathrm{C,feces}} = m_{\mathrm{C,intake}} \cdot (1 - f_{\mathrm{C}}) + w_{\mathrm{C,CP}} \cdot MP_{\mathrm{MFP}} \tag{49}$$

N excreted with urine is calculated as the residual of the animal's N balance (Eq. 1). The amount of C excreted with nitrogenous components in urine is calculated by multiplying $m_{\mathrm{N,urine}}$ with an appropriate C:N ratio. According to Dijkstra et al. (2013), the majority of N in urine (50 %–90 %) is present as urea (Dijkstra et al., 2013), which has a C:N ratio of 0.5. Around 5 % of N is present as hippuric acid with a C:N ratio of 9. The remainder are purine derivatives, creatine and creatinine, which all have C:N ratios between 1 and 1.33. This implies a plausible range of average C:N ratio in urine of about 0.95 to 1.3. For simplicity and because the amount of C excreted with urine generally makes up a small part of the C balance, we assume a C:N ratio of 1 in urine:

$$m_{\mathrm{C,urine}} = 1 \cdot m_{\mathrm{N,urine}} \tag{50}$$

C and N excreted with feces are added to the respective aboveground litter pools in LPJmL. Nitrogenous compounds in urine are assumed to be quickly degraded to ammonium. Thus, $m_{\mathrm{N,urine}}$ is added to the ammonium pool of the top soil layer in LPJmL, while $m_{\mathrm{C,urine}}$ is added to the aboveground litter pool.

## 2.10 Integration in LPJmL5.0-grazing

The new module to calculate grazing, digestion and returning C and N to the soil in form of feces and urine is implemented as a new harvest function `harvest_grass_grazing_ext_livestock` in the source code file `harvest_stand.c`. The function is called daily in the `daily_grassland` function when the grassland management option `GS_GRAZING_EXT` is set in the configuration file `lpjml.js`. The dairy cow representation is only compatible with the grassland implementation as described by Rolinski et al. (2018). The new functions of LPJmL5.0-grazing have been implemented in the LPJmL model version LPJmL5.0-tillage as described by Lutz et al. (2019).

In situations where grass biomass availability exhibits a strong seasonality, it can happen that forage intake is insufficient to fulfill daily maintenance requirements of net energy and protein at all times of the year. In order to prevent overly optimistic milk yield estimates under such circumstances, the unfulfilled daily requirements are tracked by adding the deficit to a 'buffer' (one for net energy and one for protein). When energy and protein intake are above maintenance requirements, the buffers are balanced first; milk production can only occur when both deficit buffers are zero and energy and protein intake is above maintenance requirements. To prevent that a large deficit accumulated during long deficit periods (e.g, during spin-up), impede milk production during succeeding, more productive periods, we constrain the size of the deficit buffers of energy and protein to $365 \cdot NE_\mathrm{M}$ and $365 \cdot MP_\mathrm{UP}$, respectively.

## 2.11 Modelling protocol

For the evaluation of the intake and production model alone (without interaction with LPJmL) we calculate C and N uptake and their division into C and N contained in milk, urine, feces, methane, and $CO_2$ for 1000 values of $w_\mathrm{N,CN}$ between 0.015 and 0.09 (the allowed range for grass biomass in LPJmL). We assume a body weight of 500 kg per livestock unit (LSU), unlimited grass availability, and no unfulfilled requirements of $NE$ and $MP$ from previous days.

For the simulations of the livestock model within LPJmL5.0-grazing, we also assume a body weight of 500 kg per LSU. Simulations are performed on 0.5 arc-degree resolution (about 55 km at the equator) for all global land cells except Antarctica. The model is driven by climate forcing from the GSWP3-W5E5 dataset (Kim, 2017; Cucchi et al., 2020; Lange et al., 2022), historical atmospheric deposition of $NO_3^-$ and $NH_4^+$ (Yang and Tian, 2020), and historic atmospheric $CO_2$ concentrations (Büchner and Reyer, 2022). We perform simulations with 45 different LSU densities from 0 to 4 LSU ha$^{-1}$ of grazing area; the increment between scenarios increases with increasing LSU densities from 0.01 LSU ha$^{-1}$ to 0.2 LSU ha$^{-1}$. For each LSU density setting, the model is run for 7000 years using a random permutation of years from the 1901–1931 period to bring C and N pools into equilibrium. After that, the model is run from 1901–2016 with transient climate, atmospheric deposition, and atmospheric $CO_2$ concentration.

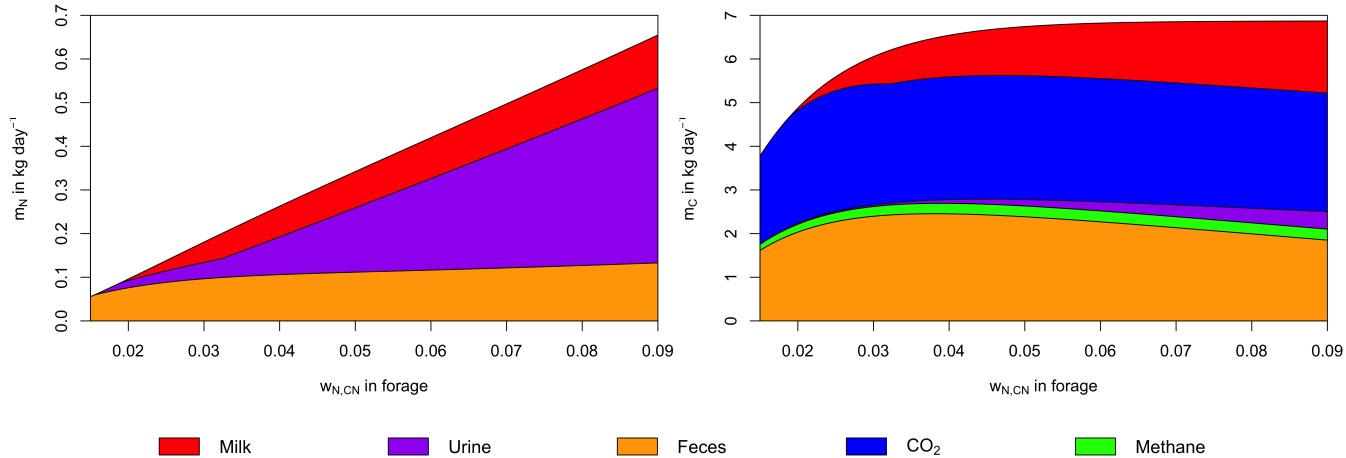

**Figure 3.** Mass balances of nitrogen and carbon as a function of $w_{N,CN}$ for a single livestock unit of 500 kg. Dry matter intake is assumed to be not limited by forage availability, and unfulfilled requirements of $NE$ and $MP$ from previous days are ignored.

## 3 Evaluation

Intake of C and N per cow as well as their division into C and N contained in milk, urine, feces, methane, and $CO_2$ changes as a function of $w_{N,CN}$ (Fig. 3). For a LSU of 500 kg, intake of C is 3.7 kg day$^{-1}$ for $w_{N,CN} = 0.015$ and increases steeply with increasing $w_{N,CN}$ until it reaches 6 kg day$^{-1}$ at $w_{N,CN} = 0.033$, which is about 90 % of the maximum daily C intake. Further increases in $w_{N,CN}$ have a comparatively small impact on C intake. In contrast, N intake increases almost linearly with $w_{N,CN}$, dominated by the increasing N content of grazed biomass. Because the digestibility of C and N increases with $w_{N,CN}$

(Fig. 2), the proportion of intake excreted with feces decreases, which results a decline in the absolute amount of C in feces for $w_{N,CN} > 0.038$. Net energy intake is sufficient to meet energy requirements for maintenance over the considered range of $w_{N,CN}$, but metabolizable protein corresponding to digested N is insufficient to meet urinary and metabolic fecal protein requirements below $w_{N,CN} = 0.019$. Thus, milk production only occurs for $w_{N,CN} > 0.019$, where it is limited by available N up to $w_{N,CN} = 0.32$ and by available energy for higher $w_{N,CN}$.

To evaluate the validity of our model, we compare the simulated partitioning of dietary N as a function of $w_{N,CN}$ to corresponding relationships from Huhtanen et al. (2008) (Fig. 4). Huhtanen et al. (2008) analyze relationships between feed properties and nitrogen utilization and partitioning in dairy cows obtained from 998 data points from 207 lactation trials. For nitrogen use efficiency (i.e., the amount of N in milk divided by total N intake), Huhtanen et al. (2008) provide nine different relationships, of which we selected the one with lowest residual mean square error (RMSE) and lowest AIC:

$$\frac{m_{N,milk}}{m_{N,intake}} = 0.627 - 33.9 \cdot \frac{w_{CP,DM}}{me} + 650 \cdot \left(\frac{w_{CP,DM}}{me}\right)^2 \tag{51}$$

For N in feces, urine, manure (feces + urine), and the fraction of urinary N in manure N, up to five different relationships are provided in Huhtanen et al. (2008), of which some contained independent variables for which we had no corresponding

estimate. From the remaining relationships, we chose the ones with lowest RMSE and AIC. N in feces is related to $DMI$ and $m_{\text{N,intake}}$:

$$m_{\text{N,feces}} = -0.021 + 0.00673 \cdot DMI + 0.101 \cdot m_{\text{N,intake}} \tag{52}$$

N in urine is related to $m_{\text{N,intake}}$, $DMI$, and $m_{\text{milk}}$:

$$m_{\text{N,urine}} = 0.04 + 0.879 \cdot m_{\text{N,intake}} - 0.009 \cdot DMI - 0.0039 \cdot m_{\text{milk}} \tag{53}$$

N in manure ($m_{\text{N,manure}} = m_{\text{N,feces}} + m_{\text{N,urine}}$) is related to $m_{\text{N,intake}}$, $DMI$, and $me$:

$$m_{\text{N,manure}} = 0.081 + 0.947 \cdot m_{\text{N,intake}} - 0.0059 \cdot DMI - 0.0059 \cdot me \tag{54}$$

The fraction of urinary N in manure N is quadratically related to $w_{\text{CP,DM}}$:

$$\frac{m_{\text{N,urine}}}{m_{\text{N,manure}}} = -0.241 + 7.11 \cdot w_{\text{CP,DM}} - 13 \cdot w_{\text{CP,DM}}^2 \tag{55}$$

To obtain the relationships with $w_{\text{N,CN}}$ shown in Fig. 4, $w_{\text{CP,DM}}$, $me$, $DMI$, $m_{\text{N,intake}}$, and $m_{\text{milk}}$ are calculated from $w_{\text{N,CN}}$ using the relationships described in section 2.

There is close agreement between relationships simulated by our model and those determined by Huhtanen et al. (2008). The partition of N intake into N in milk, N in feces, and N in urine as a function of $w_{\text{N,CN}}$ in our model is strongly supported by the relationships derived from trials by Huhtanen et al. (2008). Unfortunately, we were not able to obtain similar reference data for the evaluation of C partitioning. However, C:N ratios for all elements of the N balance are well determined or closely related to $w_{\text{N,CN}}$ in forage. Therefore, the comparison in Fig. 4 also provides an indirect validation of the C balance.

## 4   Results

With the implementation of grazing dairy systems in LPJmL, including the effect of forage quality and quantity on the uptake and partitioning of C and N, the model can now explicitly represent the effects of grazing management on land productivity and the C and N budgets.

In LPJmL5, gross primary productivity (GPP) is computed on a daily basis depending on climate, atmospheric $CO_2$ concentration, leaf area index (LAI), and plant-available water and mineral N in the soil. One direct effect of grazing in LPJmL is a reduction leaf area, which tends to reduce GPP but also autotrophic respiration. Depending on the relative strength of both effects, this can lead to an increase or a decrease in net primary productivity (i.e., the amount of carbon available for allocation to leafs and roots). Another important direct effect of grazing is the partial withdrawal of grazed N from the ecosystem (as milk) along with a transfer of the remaining N in other compartments (organic litter N and soil ammonium pools). This can result in an increase or a decrease of plant-available mineral N in soils, depending on the relative strength of the two effects. Other processes in LPJmL respond in various ways to these primary impacts of grazing and lead to changes in all stocks and flows of carbon, nitrogen, and water in the model. It should be noted, however, that the current implementation neither

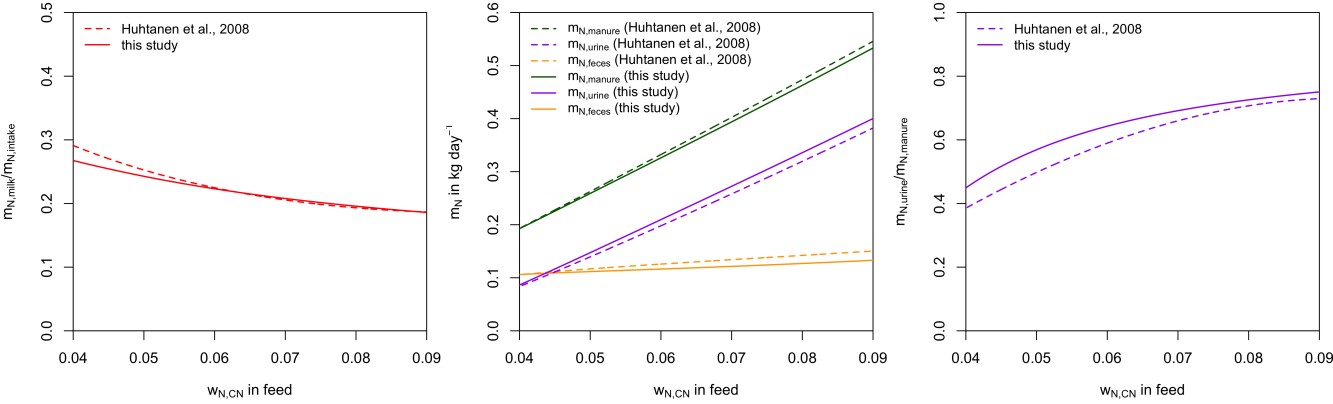

**Figure 4.** Comparison of simulated partition of dietary nitrogen as a function of $w_{\mathrm{N,CN}}$ to corresponding relationships from Huhtanen et al. (2008) obtained from experimental data. Simulation are performed assuming forage intake not limited by forage availability, and ignoring unfulfilled requirements of $NE$ and $MP$ from previous days.

considers mechanical impacts of grazing (e.g., trampling or soil compaction) nor preferential grazing of certain grass types or compartments.

As examples for site- and management specific dynamics, we show C and N balances of grassland (Fig. 5), C and N partitioning by the dairy herd (Fig. 6), and GHG emissions (Fig. 7) for Potsdam, Germany in the main text and for 3 additional sites in the Sahel in Burkina Faso (14.25°N 0.25°E), in Cordoba, Argentina (31.75°S 62.75°W), and in Riau, Indonesia (1.25°N 101.75°E) in the Appendix (Figs. A1–A9). The different sites exhibit a wide range of responses to changes in stocking density, which are caused by the interplay between direct grazing impacts and their subsequent effect on other processes in LPJmL5.0-grazing. A robust response across sites is a reduction of autotrophic respiration with increasing stocking density, which is the result of the reduction of standing biomass (Figs. 5, A1–A3). The reduction in biomass (leaf area), however, does not always lead to a reduction in GPP, which can be attributed to an increase in plant-available mineral N in the soils by grazing. An increase in N and C intake with stocking density (Figs. 6, A4–A6) is to be expected but the shape of the relationship varies among sites and is not always monotonically increasing, reflecting variations in grass availability and quality (N content). Methane emissions increase proportionally with C intake. Milk production first increases with stocking density but then decreases again, when increase in intake become smaller than increases in maintenance requirements. At the site in Indonesia (Fig. A6), this point apparently lies beyond $4\,\mathrm{LSU\,ha^{-1}}$, which is the largest stocking density used in these simulations. The N withdrawal from the ecosystem with milk production is clearly reflected in a reduction in N losses. In fact, $NO_3$ leaching and $N_2O$ emissions are usually smallest at or near the stocking density, for which highest milk yield is achieved (Figs. 5, A1–A3). This is very much in contrast to carbon sequestration in soils in response to increasing atmospheric $CO_2$ concentrations, which is typically largest for much lower stocking densities.

At medium to high stocking densities (above $1 - 2\,\mathrm{LSU\,ha^{-1}}$), total GHG emissions (expressed in $CO_2$ equivalents for a 100-year global warming potential) are usually dominated by methane emissions (Figs. 7, A7–A9). However, at low stocking densities, enhanced carbon sequestration in soils and reduced $N_2O$ emissions through grazing can offset methane emissions and in some cases even lead to a decrease in total GHG emissions (e.g., Fig. A8). Thus, there is usually a trade-off between maximizing productivity and minimizing GHG intensity (net emissions per kg of protein) at lower productivity levels. To highlight this tradeoff and to demonstrate that LPJmL5.0-grazing can be used to assess such trade-offs on a global scale, we use the set of simulations described in section 2.11 to determine for each grid cell the stocking densities corresponding to highest milk production and lowest GHG intensity (Fig. 8). The results show that GHG intensity of livestock production can indeed be negative in many locations at low or very low stocking densities but at the cost of much lower livestock productivity than possible. It is important to note that these results should be interpreted with care because dairy grazing may not be the most appropriate production system in all locations. However, the change in GHG emissions in response to grazing can be expected to be similar for other livestock grazing systems, and the level of milk production provides an indication for the level of grassland productivity, in general.

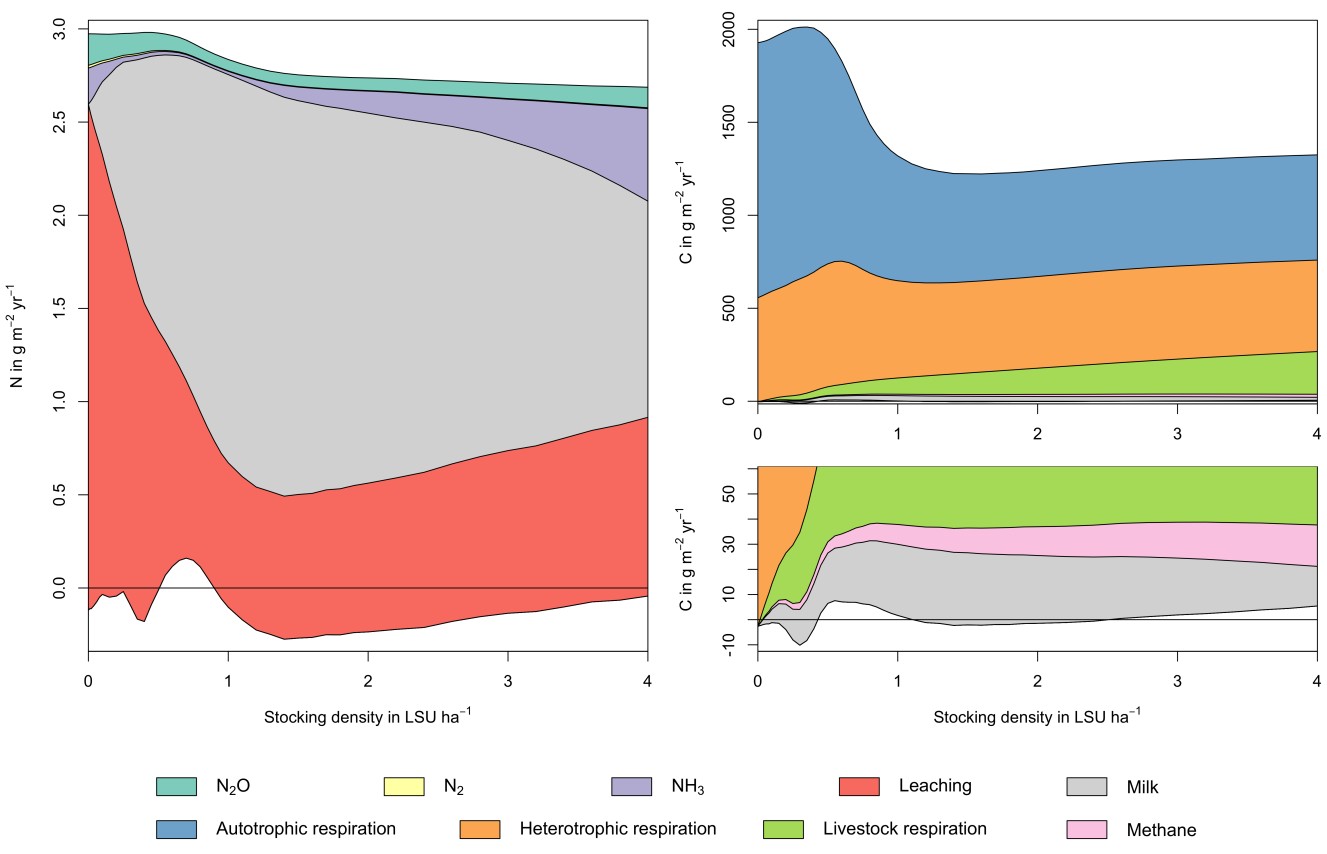

**Figure 5.** N (left) and C (right) budgets for the whole grassland system as a function of livestock density in a grid cell near Potsdam (52.25°N 13.25°E). The lower boundary represent changes in C and N soil storage. The upper boundaries represent C and N fluxes into the system: gross primary productivity for C and atmospheric deposition and biological N fixation for N. Atmospheric deposition of N at this site is $2.15\,\mathrm{g\,m^{-2}\,yr^{-1}}$. All values represent averages for 1971–2016. The lower end of the C budget (upper right) is shown in greater detail in a separate graph (lower right).

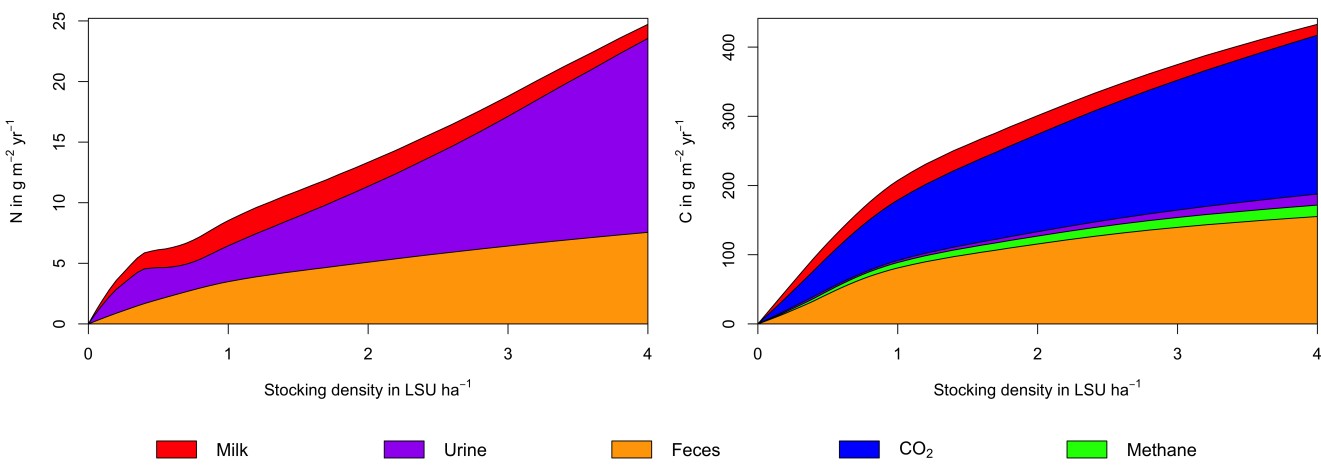

**Figure 6.** N (left) and C (right) budgets for the dairy herd as a function of livestock density in a grid cell near Potsdam (52.25°N 13.25°E). All values represent averages for 1971–2016.

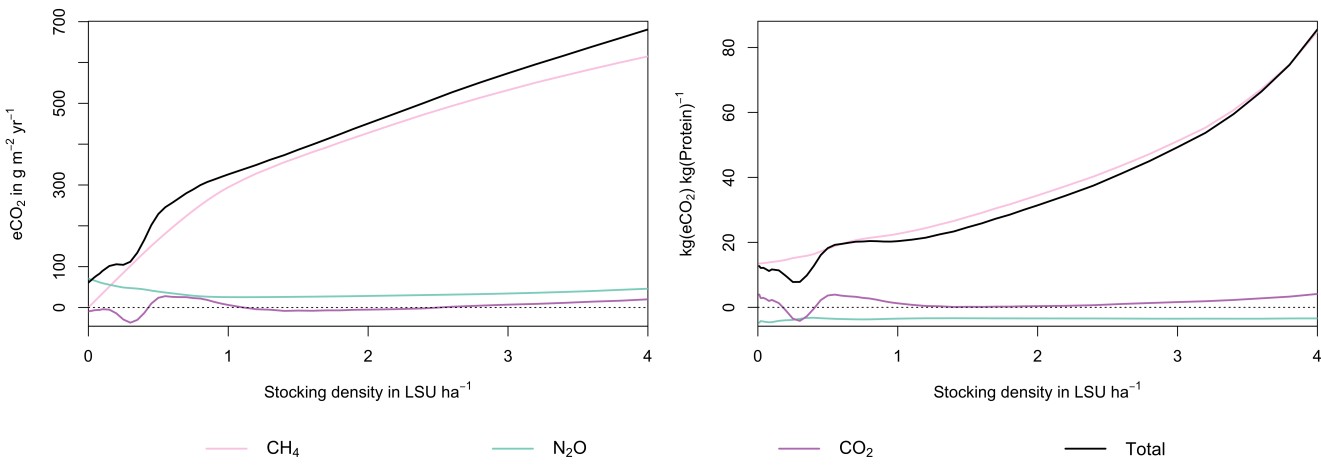

**Figure 7.** GHG emissions (left) and emission intensity (right) as a function of livestock density in a grid cell near Potsdam (52.25°N 13.25°E). Emission intensity is defined as net emissions of livestock grazing (i.e., actual emissions minus emissions for $0\,\mathrm{LSU\,ha^{-1}}$) per kg of protein produced. All values represent averages for 1971–2016.

**Highest milk production**  **Lowest emission intensity**

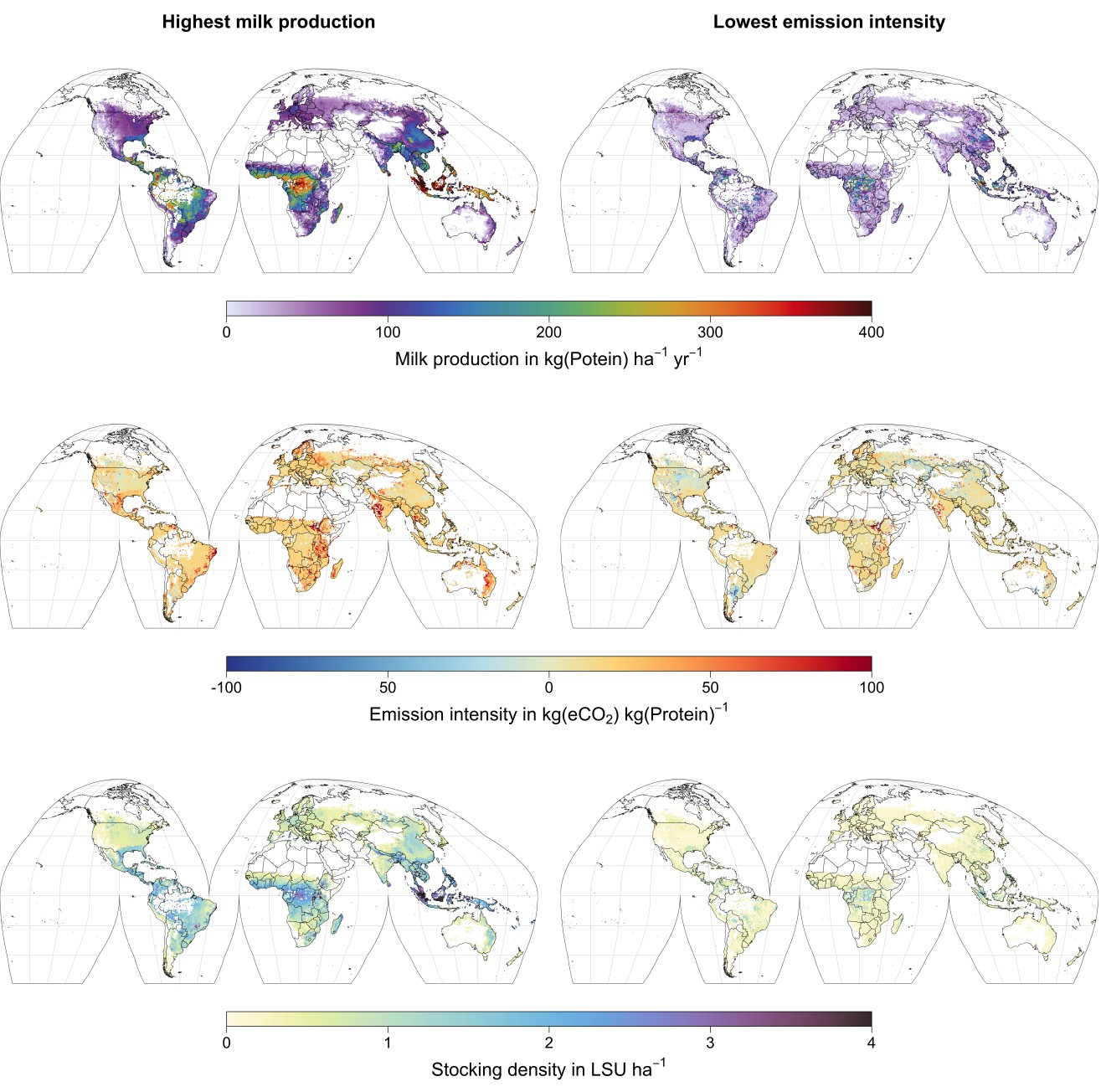

**Figure 8.** Global maps of milk production (top), and emission intensity per kg of milk protein (middle), stocking density (bottom) for the stocking densities yielding the highest milk production (left) and the lowest emission intensity per kg of milk protein (left). Only grid cells with pasture or rangeland in the year 2000 according to the HYDE3.2 dataset (Klein Goldewijk et al., 2017) are shown. All values represent averages for 1971–2016.

## 5   Discussion and conclusions

The new implementation of grazing dairy cows in LPJmL5.0-grazing greatly improves the representation of C and N cycling in grasslands under livestock grazing. The importance of forage quality, in addition to forage quantity, for the uptake and partitioning of C and N by ruminants are well known (e.g., National Research Council, 2001; Institut national de la recherche agronomique, 2007) but this has, to our knowledge, never been represented in a dynamic global vegetation model, such as LPJmL. Limitations of the model arise from the generic representation of grazing systems by grazing dairy systems, which

are not the most important systems globally (Herrero et al., 2013; Heinke et al., 2020). However, this system with continuous productivity is much easier to describe and parametrize than grazing beef cattle, which require modeling the weight gain of individual animals as well as herd dynamics. Also, the effect of supplementation with feed crops is currently not included in the model.

Despite these limitations, LPJmL5.0-grazing can be applied to assess the impacts of grazing on C and N cycles – including

carbon sequestration – for given levels of grazing intensity. For such analyses, the representation of grazing systems by grazing dairy systems is a reasonable simplification. Other possible applications include the determination of stocking densities to fulfill predefined targets (e.g., maximum productivity or lowest GHG intensity; see Fig. 8). But the results of these kinds of assessments need to be carefully interpreted in light of the generic representation by grazing dairy systems. However, such analyses can provide valuable insights on spatial variations of grassland productivity and GHG intensity, as well as the

synergies and trade-offs entailed with pasture management, which can be generalized to other forms of livestock production.

*Code and data availability.* The source code of LPJmL5.0-grazing is archived at Zenodo under https://doi.org/10.5281/zenodo.6806652 (Heinke et al., 2022).

## Appendix A: Additional site dynamics

Plots of C and N dynamics at additional sites for a marginal site in the Sahel in Burkina Faso (14.25°N 0.25°E), a site in Cordoba, Argentina (31.75°S 62.75°W), and a productive site in Riau, Indonesia (1.25°N 101.75°E) in comparison to the plots for Potsdam, Germany (52.25°N 13.25°E) in the main text.

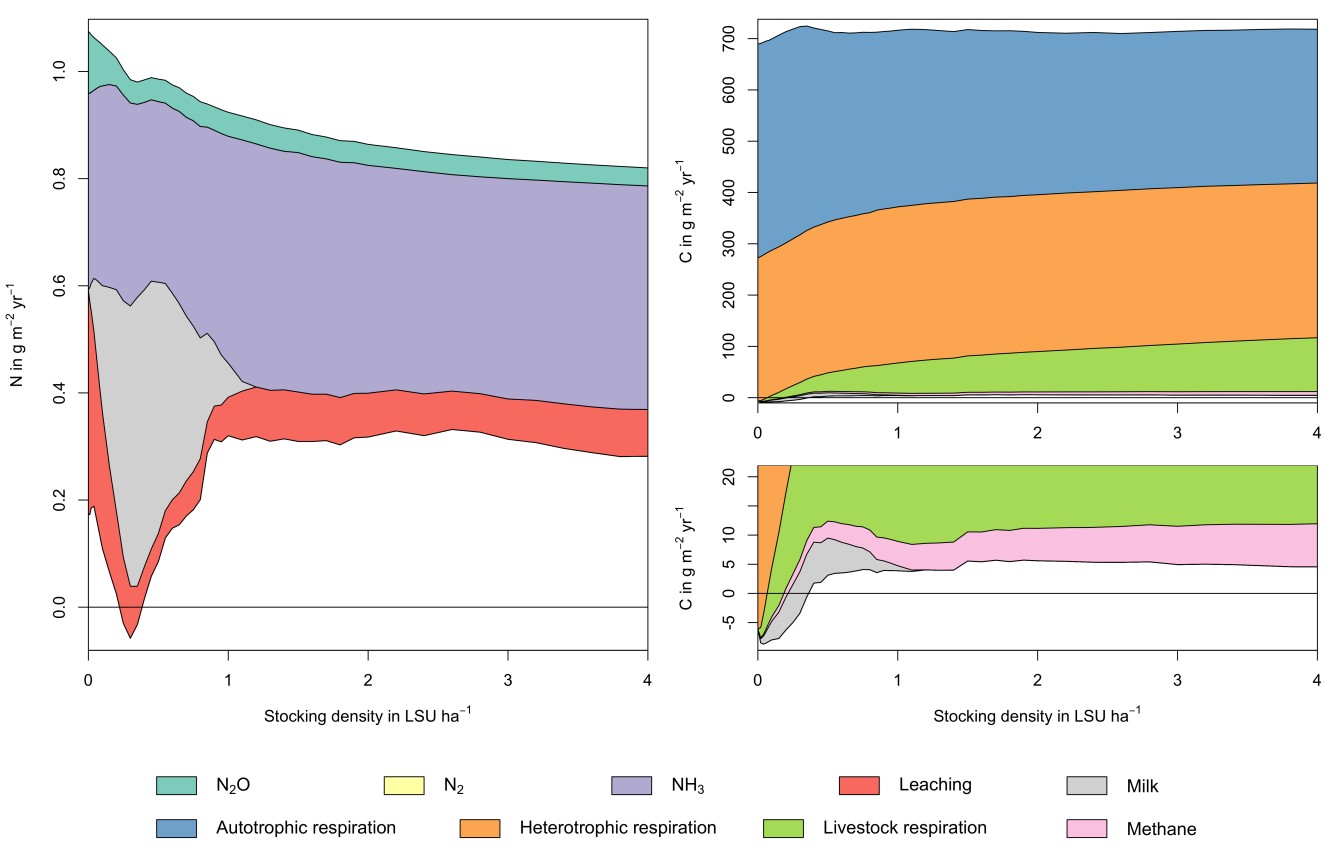

**Figure A1.** Same as Fig. 5 in the main text, but for a site in the Sahel (14.25°N 0.25°E). Atmospheric deposition of N at this site is $0.37\,\mathrm{g\,m^{-2}\,yr^{-1}}$.

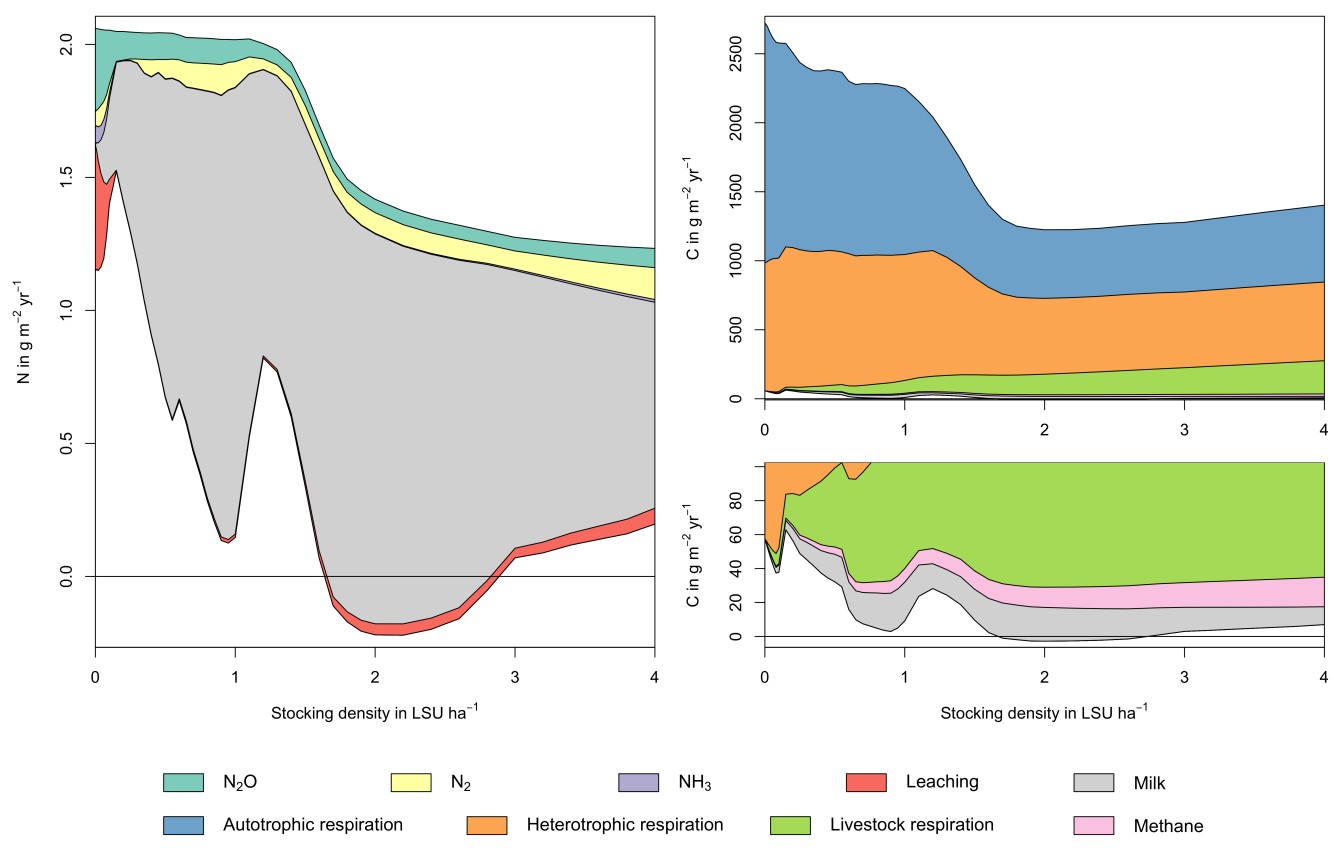

**Figure A2.** Same as Fig. 5 in the main text, but for a site in Argentina (31.75°S 62.75°W). Atmospheric deposition of N at this site is 0.40 g m$^{-2}$ yr$^{-1}$.

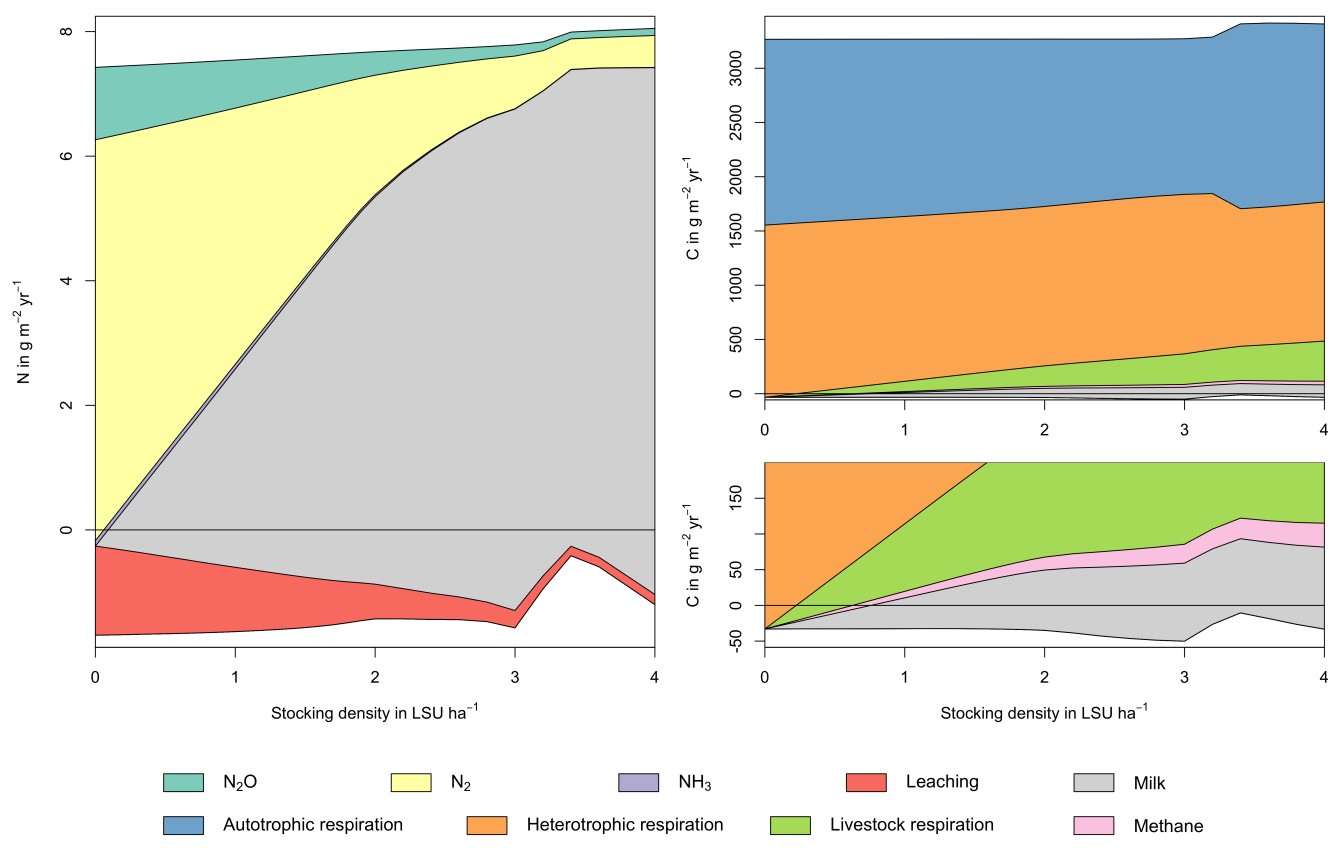

**Figure A3.** Same as Fig. 5 in the main text, but for a site in Indonesia (1.25°N 101.75°E). Atmospheric deposition of N at this site is 4.99 g m$^{-2}$ yr$^{-1}$.

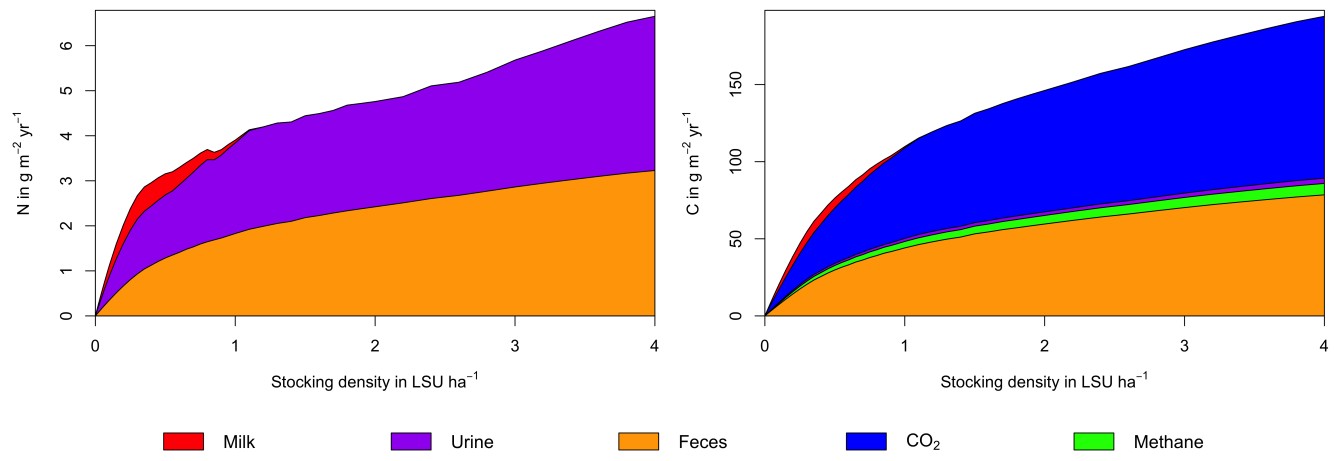

**Figure A4.** Same as Fig. 6 in the main text, but for a site in the Sahel (14.25°N 0.25°E)

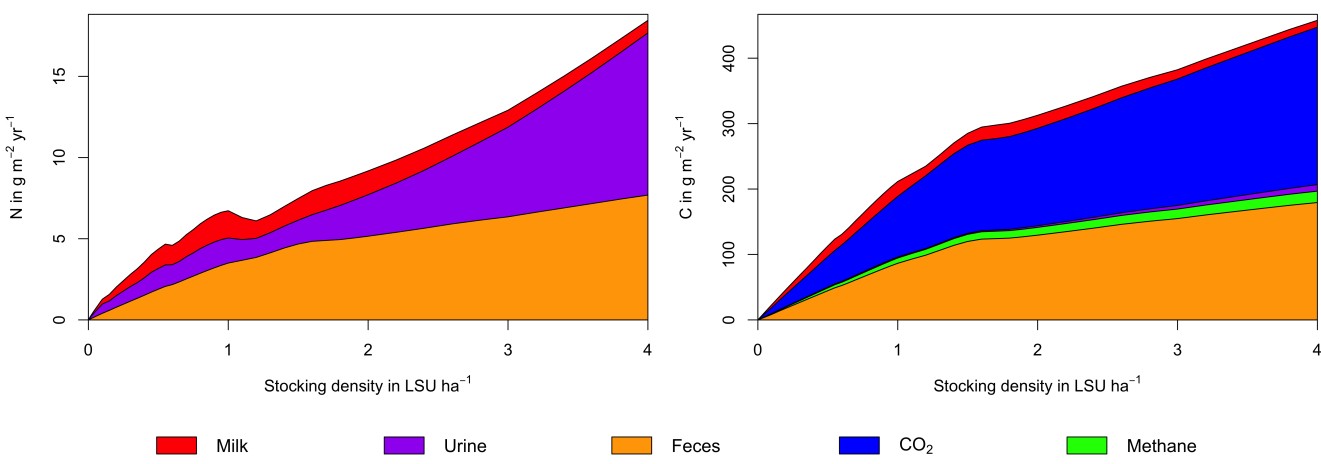

**Figure A5.** Same as Fig. 6 in the main text, but for a site in Argentina (31.75°S 62.75°W)

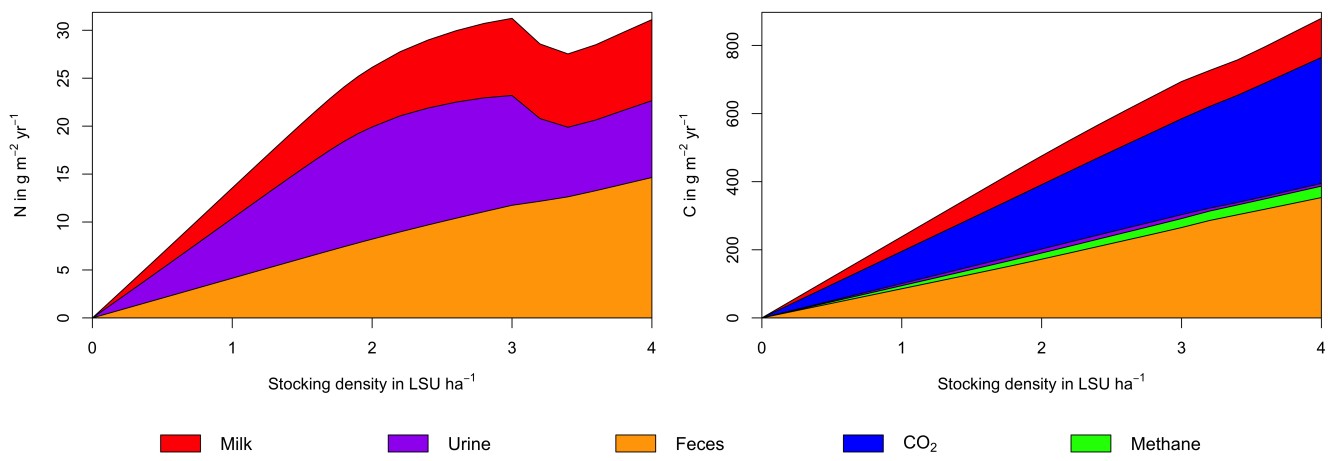

**Figure A6.** Same as Fig. 6 in the main text, but for a site in Indonesia (1.25°N 101.75°E)

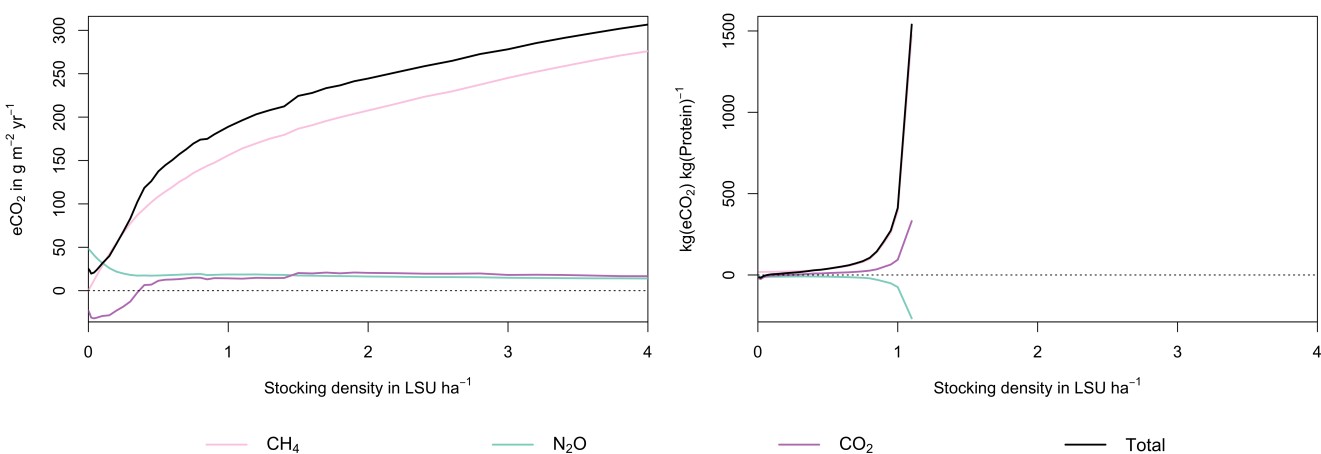

**Figure A7.** Same as Fig. 7 in the main text, but for a site in the Sahel (14.25°N 0.25°E)

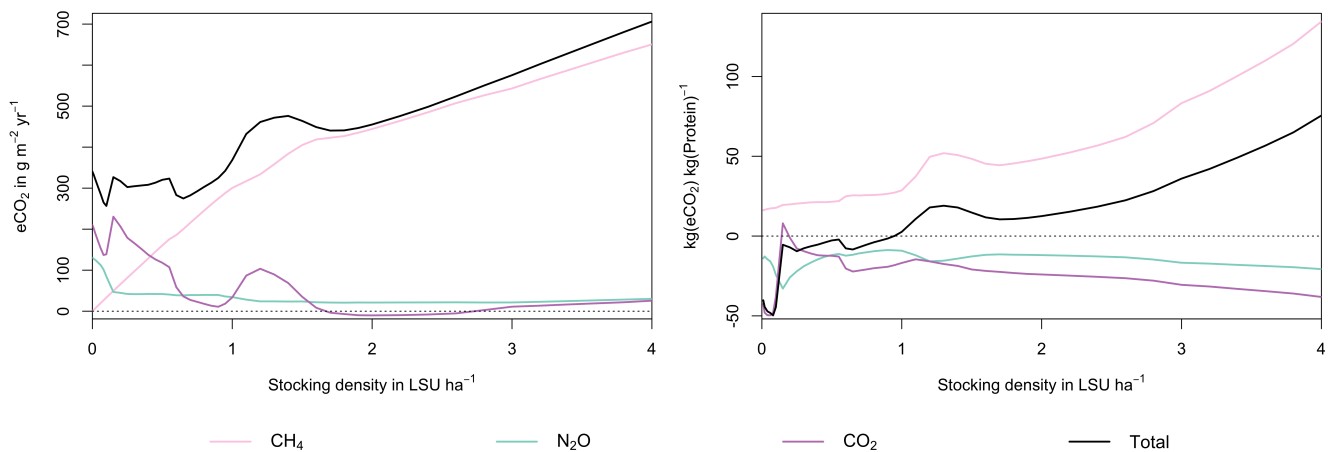

**Figure A8.** Same as Fig. 7 in the main text, but for a site in Argentina (31.75°S 62.75°W)

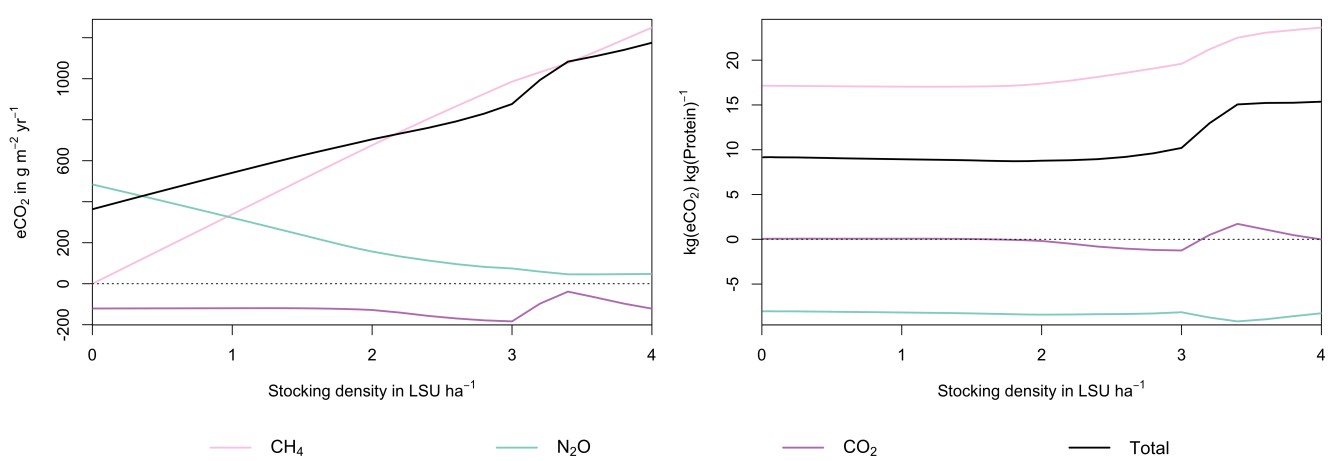

**Figure A9.** Same as Fig. 7 in the main text, but for a site in Indonesia (1.25°N 101.75°E)

**Appendix B: List of symbols**

**Table A1.** List of symbols

| Symbol | Unit | Description |
|---|---|---|
| $m_{N,urine}$ | kg day$^{-1}$ | Mass of nitrogen in urine |
| $m_{N,intake}$ | kg day$^{-1}$ | Mass of nitrogen in forage intake |
| $m_{N,feces}$ | kg day$^{-1}$ | Mass of nitrogen in feces |
| $m_{N,milk}$ | kg day$^{-1}$ | Mass of nitrogen in milk |
| $m_{C,respiration}$ | kg day$^{-1}$ | Mass of carbon used for maintenance respiration |
| $m_{C,intake}$ | kg day$^{-1}$ | Mass of carbon in forage intake |
| $m_{C,feces}$ | kg day$^{-1}$ | Mass of carbon in feces |
| $m_{C,urine}$ | kg day$^{-1}$ | Mass of carbon in urine |
| $m_{C,methane}$ | kg day$^{-1}$ | Mass of carbon in methane |
| $m_{C,milk}$ | kg day$^{-1}$ | Mass of carbon in milk |
| $w_{nNDF,DM}$ | kg kg$^{-1}$ | Weight fraction of nitrogen-adjusted neutral detergent fibre in forage dry matter |
| $w_{NDF,DM}$ | kg kg$^{-1}$ | Weight fraction of neutral detergent fibre in forage dry matter |
| $w_{NDICP,DM}$ | kg kg$^{-1}$ | Weight fraction of neutral detergent insoluble crude protein in forage dry matter |
| $w_{CP,DM}$ | kg kg$^{-1}$ | Weight fraction of crude protein in forage dry matter |
| $w_{NFC,DM}$ | kg kg$^{-1}$ | Weight fraction of non-fiber carbohydrates in forage dry matter |
| $w_{EE,DM}$ | kg kg$^{-1}$ | Weight fraction of ether extract in forage dry matter |
| $w_{A,DM}$ | kg kg$^{-1}$ | Weight fraction of ash in forage dry matter |
| $w_{FA,DM}$ | kg kg$^{-1}$ | Weight fraction of fatty acids in forage dry matter |
| $w_{L,DM}$ | kg kg$^{-1}$ | Weight fraction of lignin in forage dry matter |
| $w_{ADICP,DM}$ | kg kg$^{-1}$ | Weight fraction of acid detergent insoluble crude protein in forage dry matter |
| $w_{C,CP}$ | kg kg$^{-1}$ | Weight fraction of carbon in crude protein |
| $w_{N,CP}$ | kg kg$^{-1}$ | Weight fraction of nitrogen in crude protein |
| $w_{C,CHO}$ | kg kg$^{-1}$ | Weight fraction of carbon in carbohydrates |
| $w_{C,L}$ | kg kg$^{-1}$ | Weight fraction of carbon in lignin |
| $w_{C,FA}$ | kg kg$^{-1}$ | Weight fraction of carbon in fatty acids |
| $w_{C,DM}$ | kg kg$^{-1}$ | Weight fraction of carbon in forage dry matter |
| $w_{N,DM}$ | kg kg$^{-1}$ | Weight fraction of nitrogen in forage dry matter |
| $w_{N,CN}$ | kg kg$^{-1}$ | Weight fraction of nitrogen in sum of carbon and nitrogen |
| $d_{NFC}$ | kg kg$^{-1}$ | Weight fractions of digestible non-fiber carbohydrates in total dry matter |
| $d_{CP}$ | kg kg$^{-1}$ | Weight fractions of digestible crude protein in total dry matter |
| $d_{FA}$ | kg kg$^{-1}$ | Weight fractions of digestible fatty acids in total dry matter |

Continued . . .

| Symbol | Unit | Description |
|---|---|---|
| $d_{\text{nNDF}}$ | $\text{kg kg}^{-1}$ | Weight fractions of digestible nitrogen-adjusted neutral detergent fibre in total dry matter |
| $d_{\text{ADICP}}$ | $\text{kg kg}^{-1}$ | Weight fractions of digestible acid detergent insoluble crude protein in total dry matter |
| $f_{\text{C}}$ | $\text{kg kg}^{-1}$ | Digestible fraction of carbon in total dry matter |
| $f_{\text{N}}$ | $\text{kg kg}^{-1}$ | Digestible fraction of nitrogen in total dry matter |
| $BW$ | kg | Body weight |
| $DMI_{max}$ | $\text{kg day}^{-1}$ | Dry matter intake capacity |
| $DMI$ | $\text{kg day}^{-1}$ | Dry matter intake |
| $q$ | - | Slope parameter in Eq. 27 |
| $K$ | - | Position parameter in Eq. 27 |
| $de$ | $\text{Mcal kg}^{-1}$ | Digestible energy in forage dry matter |
| $me$ | $\text{Mcal kg}^{-1}$ | Metabolizable energy in forage dry matter |
| $ne$ | $\text{Mcal kg}^{-1}$ | Net energy in forage dry matter |
| $NE_{\text{M}}$ | $\text{Mcal day}^{-1}$ | Net energy requirements for maintenance |
| $NE_{\text{milk}}$ | $\text{Mcal kg}^{-1}$ | Net energy requirements for milk production |
| $MP_{\text{avl}}$ | $\text{kg day}^{-1}$ | Available metabolizable protein |
| $MP_{\text{UP}}$ | $\text{kg day}^{-1}$ | Metabolizable protein requirements for urinary protein |
| $MP_{\text{MFP}}$ | $\text{kg day}^{-1}$ | Metabolizable protein requirements for metabolic fecal protein |
| $MP_{\text{milk}}$ | $\text{kg kg}^{-1}$ | Metabolizable protein requirements for milk production |
| $m_{\text{milk,NE}}$ | $\text{kg day}^{-1}$ | Potential milk production from net energy |
| $m_{\text{milk,MP}}$ | $\text{kg day}^{-1}$ | Potential milk production from metabolizable protein |
| $m_{\text{milk}}$ | $\text{kg day}^{-1}$ | Milk production |
| $w_{\text{fat,milk}}$ | $\text{kg kg}^{-1}$ | Weight fractions of fat in milk |
| $w_{\text{protein,milk}}$ | $\text{kg kg}^{-1}$ | Weight fractions of protein in milk |
| $w_{\text{CHO,milk}}$ | $\text{kg kg}^{-1}$ | Weight fractions of carbohydrates in milk |
| $w_{\text{C,milk}}$ | $\text{kg kg}^{-1}$ | Weight fractions of carbon in milk |
| $w_{\text{N,milk}}$ | $\text{kg kg}^{-1}$ | Weight fractions of nitrogen in milk |
| $m_{\text{methane}}$ | $\text{kg day}^{-1}$ | Methane production |

*Author contributions.* JH developed the model, perfomed simulations, and analysed results. JH, SR, and CM wrote the paper.

*Competing interests.* The authors declare that there are no competing interests.

*Acknowledgements.* JH acknowledges funding from the EXIMO project (01LP1903D) funded through the German Ministry for Education and Research (BMBF). SR acknowledges funding from the projects AGrEc (01DG21039) and CLIMASTEPPE (01DJ18012), both funded through the BMBF.

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
