# Peer review of "Modeling the role of livestock grazing in C and N cycling in grasslands with LPJmL5.0-grazing"

_Geoscientific Model Development, 2022_

## Author Response (AR1)

*We thank all three reviewers for their detailed and constructive comments. Below, we provide point-by-point responses to all specific comments. Reviewers' comments are in regular font, our responses are in italics. Line numbers referring to changes in the manuscript are **line numbers in the manuscript with tracked changes**. We have indexed all major comments to facilitate cross-referencing.*

**Anonymous Referee #1**

The authors developed a module simulating C and N dynamics of dairy cow grazing. The module accounts for the C:N ratios in grazed biomass, and its impacts on dry matter intake by cows and the C and N partitioning between milk, urine, feces, methane and respiration CO2. The module was established based on (mostly empirical) demand and supply equations for metabolizable and digestible energy and protein, empirical functions between characteristics of feeds, and data from feed database and literature. The module was then implemented in to a dynamic global vegetation model LPJmL to simulate the C and N dynamics of a grassland ecosystem under dairy cow grazing with different grazing density, including dynamics in vegetation, soil, and livestock as well as the ecosystem GHG emissions. The module is well constructed, the equations are justifiable, and the validations of the module and the model over four grassland sites are in general well conducted. The manuscript is well written and deserve a publication after revision.

*We thank the reviewer for their positive overall evaluation.*

I have a few suggestions for consideration during the revision.

[R1.C1] Section 2 "Model description" is long and with many detail, which is necessary for a model description paper and is appreciated. But a schematic diagram linking all components of the module and the equations should be added to help readers get an overview of the module/model. In addition, a look-up table listing all abbreviations, variables and parameters used in the manuscript would be necessary.

*We appreciate these suggestions. Similar suggestions have been made by reviewer #3 in their general comments, as well. We have added a schematic overview of the model's structure and a list with all mathematical symbols used in the revised manuscript (Fig. 1 and Appendix A2, respectively). In addition, we include an R script with a fully functional implementation of the livestock module as supplementary material.*

[R1.C2] Given the fact that 1) this module only focus on dairy cow grazing over grassland, 2) the manuscript is mainly describing the module and the site application to infer the model's performance, and 3) cow grazing is one of the many herding systems in the world (actually not a dominantly widely spread one, especially over vast semi-arid rangeland), it is not needed and appropriate to conduct global simulations, which is not informative at all and out of the scope of this study. In fact, the authors did not pay much attention on the global results too.

*We concur that the maps have been given too little attention in the text. However, we disagree that the maps have no use. LPJmL is routinely applied to conduct global scale simulations, and to demonstrate that the model can be applied globally to give reasonable results is important. The maps are derived from the same simulations that*

*were used to create the site-specific figures and show results for two different livestock densities related to two selected points of interest over the range of tested livestock densities (maximum production and minimum GHG intensity). We have revised and extended the results section to clarify the link between the site-specific figures and the maps (e.g., L473-476), and to highlight the stylized nature of these two scenarios (L478-481).*

[R1.C3] The title and the abstract should be revised to reflect the main point of this study. It is dairy cow grazing rather than livestock grazing.

*We use dairy cows as a generic representation of grazing livestock to model the effect of grazing on grassland C and N cycling. We have revised the abstract to highlight this (L4) and to clarify the applicability of such a model (L11-15). Furthermore, we have revised the results and discussion sections to match this framing and to highlight the limited interpretability of productivity-related results (e.g., L478-481, L491-192, and L501-505).*

Minor remarks:

L76: It is not there what is the wFA in Eq. 6 used for.

*wFA is used in Eq. 15c to calculate dFA, which is required in Eqs. 17 and 19 to determine carbon content and digestible energy, respectively, of forage dry matter.*

Fig. 7 unit for upper panel should be kg(Protein) ha-1 yr-1, while the figures could be deleted to make the manuscript more focused.

*We have corrected this typo in the revised manuscript. However, we have kept this figure as explained in our response to [R1.C2].*

**Anonymous Referee #2**

This paper presents a module for the LPJmL5.0 model to represent the role of grazing lactating dairy cows in N and C cycling in grassland. The new module accounts for feed quality on animal intake and partitioning of C and N between milk, feces, urine, respiration, etc.

The main contribution of this paper is clearly the description of feed quality parameters and its relation to animal intake and metabolism. The section "Model description" is very well written and carefully goes through the main considerations behind the model. This part of the paper has a considerable reuse value for the agro-food system modeling community above and beyond the specifics of the LPJmL model.

Apart from a well-written method section in the paper, the authors have documented their methods in the form of full source code, which in principle ensures full reproducibility of all the calculations. Having no previous experience with the inner workings of LPJmL, I have (within the scope of this review assignment) not been able to assess the quality or completeness of the source code, and so I cannot comment on the code except that it is published and appears at a quick glance to be complete. I applaud the effort to publish the source code and I encourage the authors and colleagues to continue on this path with future work.

A few critical remarks follow, none of which challenge the basic contribution of the paper, but several of which I would strongly suggest to address to ensure that the paper and the applicability of the model is not misunderstood.

*Thank you for the very positive overall evaluation!*

[R2.C1] First and foremost, I am a little confused about whether the authors see their module as specifically representing lactating dairy cows or as a generic description of grazing livestock using the lactating dairy cow as a representative animal. The formulation of the title and introduction (e.g., line 29), as well as most of the abstract, led me to believe the latter proposition; but the discussion/conclusion section (lines 412-417) rather led me to think that the authors see the model as limited to lactating dairy cows. I would strongly suggest that the model description section (Section 2) be amended (preferably at the start of the section) with a clarifying paragraph on this, and in addition that the discussion and conclusions be elaborated on the limitations of the model. Indeed, as the authors note (lines 412-413) "grazing dairy systems are not the most important systems globally", and it is therefore crucial to clarify the applicability and limitations of the model to represent grazing livestock.

*We see the described dichotomy in the framing and agree that it causes confusion. The primary motivation for this implementation was the improvement of carbon and nitrogen cycling in grasslands, for which dairy cows were chosen as a representative animal. However, the implementation also provides a measure of productivity and GHG intensity of ruminant production on global grasslands that is comparable across sites, despite the fact that dairy farming is not the most likely form of livestock rearing in many cases. We have revised the abstract to highlight the generic representation of grazing by dairy cows (L4) and to clarify the applicability of such a model (L11-15). Furthermore, we have revised the results and discussion sections to match this framing and to highlight the limited interpretability of productivity-related results (e.g., L478-481, L491-192, and L501-505).*

[R2.C2] Second, I find that the framing of the paper is a bit overreaching considering that the new module is concerned only with feed intake and digestibility for lactating dairy cows. The paper at a first glance (title and partly the abstract) appears to cover "livestock grazing" in general. Moreover, the mention in the abstract (lines 9-12) of "effects of management and climate change" and "Optimal stocking densities" and "best NUE, or highest C sequestration" led me to expect a paper dealing also with the effects of livestock grazing management on crop growth. As I understand the model, however, the effect of grazing livestock is only accounted for in terms of animals' grazing intake and partitioning of C and N. Other effects of grazing livestock, such as the effects of defoliation and trampling on botanical composition, crop growth, soil compaction, etc., are not accounted for. The results section (Section 4) shows some results of gross primary productivity (Figure 4) but I cannot easily see how these were derived and so I guess that is from other parts LPJmL model fed with the N and C inputs of the grazing dairy cows, without accounting for other (non-nutrient) effects of grazing livestock on grassland productivity. This is an understandable and completely acceptable limitation of the present study, so the point of my remark is not to complain about the limitation but rather about the mismatch between my expectations of the paper based on the title and abstract and the wide net cast by the introduction.

*We have removed the statements in question in the abstract and have sharpened the framing towards impacts on C and N cycle in the revised manuscript. We have added a paragraph explaining the interaction of the new livestock module with other processes in LPJmL5 (L427-436), also highlighting missing effects (e.g., as soil compaction and trampling). We have extensively revised and extended the results section to further highlight the effects of the nutrient-related interactions and to better reflect the scope of the implementation.*

[R2.C3] On the same note, I find that the evaluation section (Section 3) makes a good and convincing validation of the lactating dairy cows' partitioning of N; but it says nothing about grassland productivity.

My suggestion therefore also on this point is to amend the abstract, introduction and discussion/conclusion sections with a clarification of the scope and limitations of the present study. Also the paper's title could perhaps be somehow adjusted to more precisely reflect the scope of the study.

*We have revised and extended abstract, results, and discussion sections to highlight that the paper is primarily about grazing impacts on C and N cycles (L1-2, L424-426, L483-486, 499-501), that grazing livestock is represented by dairy cows (L4, L33-34, L492-496), and that this limits interpretability of productivity-related estimates (L478-479, 491-492, L502-503). However, we note that the model can provide insights about variations in grassland productivity in space and time and into trade-offs between livestock productivity and environmental impacts (L14-15, 479-481, 503-505).*

[R2.C4] Third, from a practical usage perspective I wonder how the model is prepared to handle grazing only during part of the day and/or the year and supplemental feeding in houses and/or on the pasture. These are common practices in many regions that strongly affect the partitioning of N and C, and therefore I would see little practical use of the model unless it can flexibly handle such variants. I would suggest that the authors at least elaborate on this in the paper; and perhaps they might also be interested in explicitly representing the possibility in the model.

*Different forms of grazing management (e.g., continuous grazing, rotational grazing, mowing) have previously been implemented in LPJmL (Rolinski et al., 2018; cited in the manuscript) and can be combined with the implementation presented here. However, we decided not to explore this dimension in the manuscript and keep the focus on the new livestock module itself and how it performs under continuous grazing (the default) in LPJmL. Supplementation with feed crops is planned to be included in a future version of the model. We have revised the manuscript to make it clear that cows only consume the grass they graze (e.g., by using 'forage' instead for 'feed' and by clarifying the framing), and we explicitly note that the omission of supplemental feeding is a limitation of the model (L479-498).*

Minor line-specific comments:

L55: The overview of the section skips subsections 2.1 to 2.4.

*These sections are referred to in the previous sentences (L54-59).*

L79: Perhaps say "20 commonly occurring proteinogenic amino acids" or "The 20 most common amino acids have molar weights between ..." or similar, to avoid any confusion about the total number of naturally occurring amino acids.

*We have changed the sentence as proposed.*

L249: "urine of methane" --- probably a typo; should it say "urine or methane"?

*Yes, this is a typo. We have corrected it in the revised manuscript.*

L260: "10% activity allowance" --- is this a reasonable estimate for grazing lactating dairy cows? I imagine that the NRC 2001 data are based on lactating dairy cows mostly staying inside eating high-quality feed and not walking around very much on a pasture.

*We agree that this is probably too low for free-ranging cattle. We have added a comment on this in the revised manuscript (L290-292). An estimation of energy requirements for activity in relation to feed availability and other factors is planned for a future version of the model.*

L299: 18.4 MJ/kg gross energy content of "feed" --- strikes me as unspecific and also poorly motivated compared to the high level of detail in most of the model. Should not the gross energy content of the feed (i.e., whatever grows on the grassland, I suppose) be an endogenous variable of the model?

*The main uncertainty in this equation is the assumed emission factor (constant 6.5 %). Of course, it would be possible to use gross energy estimated from feed composition here. But it would merely pretend an improvement in accuracy. We have added a comment on this in the revised manuscript (L334-336).*

L337 (and elsewhere): The acronym LSU probably means livestock unit? Please spell out the meaning.

*We have added the explanation of LSU at its first occurrence in the revised manuscript (L374-375).*

L342-343: Atmospheric deposition of N, I suppose, is included as an N input into the grassland with an effect on primary production. What about biological fixation, which is likely in most grasslands at least as big or bigger, and can also vary strongly with pedoclimatic conditions, grazing management, seeding, nutrient inputs, etc.

*Biological N fixation is calculated endogenously in LPJmL. It was mentioned in the caption of Fig. 4 (Fig. 5 in the revised manuscript) that N fluxes into grasslands comprise atmospheric deposition and biological N fixation (BNF). We have added the rate of atmospheric deposition at each site in the captions of Figs. 5 and A1-A3 so that changes in BNF with stocking density can be evaluated by the reader.*

**Anonymous Referee #3**

General comments

This paper presents the development of a dynamic global vegetation model called LPJml5.0-grazing, which now includes a simplified representation of the grazing of lactating dairy cows and the fate of the nitrogen and carbon they ingest. Nitrogen is thus partitioned between milk, urine and feces and carbon between milk, urine, feces, respiration and methane. The intake of carbon and nitrogen takes into account the chemical composition of the ingested forage and the digestibility and abundance of these constituents. This refinement is what makes this model original. The model was not evaluated by comparing it to observed situations, which prevents the calculation of statistical error criteria, but by comparing it to an empirical modeling of the fate of ingested nitrogen and carbon proposed by Huhtanen et al. A simulation of the partitioning of ingested carbon and nitrogen is done at several sites and an illustration is given of partitions as a function of animal density that has been optimized to meet a milk production or environmental objective. I recommend that the authors make some revisions to their paper to make it easier to understand and to provide more information on the model itself and its domain of validity. In particular, the reading of the paper is not always easy and some points seem to me to be clarified. The descriptive part of the model would be easier to follow if if a scheme linking the different main variables and a lexicon of the different variables containing their abbreviation, their definition and their unit were given to the reader. The discussion of the results is coupled with the conclusion. Few elements are discussed. I think we should go back more to the area of validity of the domain, what is taken into account and what is not.

*We thank the reviewer for their constructive comments. As mentioned in our response to [R1.C1], we have added a schematic overview of the model's structure and a list with all mathematical symbols used in the revised manuscript (Fig. 1 and Appendix A2, respectively). We also include an R script with a fully functional implementation of the livestock module as supplementary material. Furthermore, we have revised and extended the results and discussion/conclusion sections to include a more extensive description of model behavior and to clarify the intended application of the model and its limitations.*

Specific comments

[R3.C1] There is mention of feed or forage. What kind of feed is really simulated for the cows? Are the cows only fed with the grass they graze? Or does the model offer the possibility to take into account a supplementation of the cows during the grazing period? From reading paragraph L327 I have the impression that cows are only well fed with fresh grass and that this explains why they can find themselves in a situation where this grass is no longer sufficient to cover their needs. If it is indeed grazed grass please use grass instead of feed in the whole paper.

Also, does the chemical composition of the grass/feed ingested by the animals vary during a simulation or is it assumed to be identical during the simulation?

These points seem to me important to clarify and discuss at the end. Because if the cows are only fed with grass and that the quality of the grass remains the same during the simulation these are limits of the model.

*In the present version of the model, cows receive only the grass they graze. We have replaced most instances of 'feed' by 'forage' in the revised manuscript to avoid confusion, and we state the omission of supplementation with feed crops as a limitation of the model (L479-498). However, the composition (C:N ratio) of grazed biomass changes dynamically in LPJmL, which is the very reason why the relationships between feed properties and the mass fraction of N in C and N were derived (section 2.4, Fig. 2). We have added a note that this fraction varies in LPJmL5.0-grazing (L374) and that C and N pools respond dynamically to grazing (L433-435).*

[R3.C2] Lines 151 to 157. I have the impression that there is an error at this level. Does the variable d correspond to a digestible fraction or to a digestible quantity? If it is a fraction, shouldn't the variables for mass wNFC wCP etc. be in the equations? When using these equations (15a) to (15d) as written in the paper in equation (17) the biomasses in the numerator are squared.

*Weight fractions of nutrients and digestible nutrients (e.g., w_NFC and d_NFC), are both in kg(nutrient) per kg(dry matter). In equation 17, digestible nutrients are multiplied by their respective carbon fraction (in kg(carbon) per kg(nutrient)), yielding digestible carbon in kg(carbon) per kg(dry matter). Division by total carbon in dry matter (in kg(carbon) per kg(dry matter)) gives fraction of digestible carbon (in kg(carbon) per kg(carbon)). This should be easier to follow with the table of mathematical symbols included as Appendix A2 in the revised manuscript.*

[R3.C3] L210. The statement "none of them has been explicitly developed for lactating cows" is false. I encourage the authors to read the publications for instance of Delagarde Remy et al. Concerning the GrazeIn model and modify this sentence to reflect the fact that such relationships already exist.

*We are thankful for this comment, which motivated us to have another look at the INRA fill unit system. Unfortunately, most of the documentation of the INRA system is in French, which makes it less accessible than the NRC system, on which our livestock module is based. However, the main problem is that the calculation of feed intake in the INRA system requires iteration between multiple equations of the INRA system, which makes it incompatible with the NRC system. We have added a note on this in the revised manuscript, and we will continue to investigate the usefulness of the INRA system for future model development.*

[R3.C4] L337. How was the interval between 0 and 0.10 determined?

*We have changed the interval to 0.015-0.09, which is the allowed range for grass biomass in LPJmL (L374).*

[R3.C5] L340. What resolution in km or ha does it correspond to?

*About 55 km at the equator. We added this in the revised manuscript (L377).*

[R3.C6] L344. Can you specify the unit of animal densiy? is it per hectare of useful agricultural area? Per hectare of main forage area?

*Stocking density is given per hectare of grazing area. We have clarified this in the revised manuscript (L380).*

[R3.C7] Lines 390-393. The descriptive part of the model does not explain how these animal density optimizations are achieved to achieve production or environmental objectives. Please add elements so that the reader can understand how these optimizations are carried out.

*There was no true optimization carried out. Instead, livestock densities (and corresponding variables) for the two points of interest were determined from the same set of simulations that was used to create the site-specific figures (Figs. 5-7 and Figs. A1-A9). We have removed the misleading term 'Optimized' from the maps in Fig. 8 and have clarified the relationship between the maps and the site-specific figures in the revised manuscript (L474-475).*

[R3.C8] Figure 4. How do you explain that the share of leached nitrogen is so important at low animal densities?

*The biophysical conditions at this site (soil and climate) cause nitrogen from atmospheric deposition and biological N-fixation to be lost primarily by leaching. As livestock density increases, an increasing share of nitrogen is withdrawn from the system with the produced milk. We have added an explanation of this interaction in the revised manuscript (L460-462).*

Technical corrrections

L38. Write cows instead of cattle

*Changed as suggested (L42)*

L52. Please add « respectively » after wN,CN or move (wC,DM) after " the weight fraction of feed in dry matter"

*Changed as suggested (L56)*

L105. Write « fractions » instead of « factions »

*Changed as suggested (L115)*

L197. RSE instead of RSA

*Changed as suggested (L220)*

L231. Write biomass availability instaed of biomass viability

*Changed as suggested (L260)*

L262. MP used to abreviate «Metabolizable protein » can be confused with « Milk production »

*MP is commonly used to abbreviate metabolizable protein in the NRC system and animal nutrition literature. We believe that the risk of confusion is minimal with the table of symbols included as Appendix A2 in the revised manuscript.*

L307. Write mC,intake instead of mC,in

*Changed as suggested (L344)*

L309. Write mC,urine instead of mC,feces

*Changed as suggested (L346)*

L326 If accurate, specify « grass » biomass availability

*Changed as suggested (L363)*

L325 If accurate, specify «milk » production

*There is no occurrence of 'production' in L325. In case this comment refers to "feed intake and production model" (L335 in the original manuscript), we found it preferable to simplify this to "intake and production model" (L372).*

Figure 2 (and others). using « WN in feed » leads to confusion. why not use « WN,CN »?

*Changed as suggested (Fig. 3)*

Figure 3. Write « as a function of WN,CN »

*Changed as suggested (Fig. 4)*